# Conditional Local Importance by Quantile Expectations

**Kelvyn K. Bladen**                                         *kelvyn.bladen@usu.edu*
*Department of Mathematics & Statistics*
*Utah State University*

**Adele Cutler**                                              *adele.cutler@usu.edu*
*Department of Mathematics & Statistics*
*Utah State University*

**D. Richard Cutler**                                        *richard.cutler@usu.edu*
*Department of Mathematics & Statistics*
*Utah State University*

**Kevin R. Moon**                                             *kevin.moon@usu.edu*
*Department of Mathematics & Statistics*
*Utah State University*

**Reviewed on OpenReview:** *https://openreview.net/forum?id=gsuZFPDRqE*

## Abstract

Global variable importance measures are commonly used to interpret the results of machine learning models. Local variable importance techniques assess how variables contribute to individual observations. Current, popular methods, including LIME and SHAP, provide useful measures of feature contribution in the prediction space, while leaving opportunities for improved characterization of local structure in the model loss space. Additionally, they are not natively adapted for multi-class classification problems. We propose a new model-agnostic method for calculating local variable importance, CLIQUE, that highlights locally dependent relationships, provides improved stability over permutation-based methods, and can be directly applied to multi-class classification problems. Simulated and real-world examples show that CLIQUE emphasizes locally dependent information, captures interaction behavior beyond what can be evaluated by correlations, and assigns zero importance in regions where the response is invariant to changes in variables.

## 1 Introduction

Variable importance measures are essential for interpreting machine learning models, as they evaluate the contribution of each feature to the model's performance or predictions. In particular, global variable importance methods quantify a feature's overall influence across the entire dataset and have been extensively researched. Breiman (2001) introduced a global permutation technique (Permute-and-Predict or PaP) for an internal validation set within Random Forests, which was later generalized to model-agnostic approaches (Model-Reliance) (Fisher et al., 2019). Other global variable importance methods include Leave-One-Covariate-Out (LOCO) (Lei et al., 2018; Hooker et al., 2021; Bladen & Cutler, 2024), the Knockoff Filter (Barber & Candès, 2015; Candes et al., 2018), and using the spread of predictions in Partial Dependence Plots (PDPs) (Greenwell, 2017; Greenwell et al., 2018).

Local variable importance seeks to measure the relative contribution of each feature to the performance or prediction of individual observations. Existing methods include Individual Conditional Expectations (ICE) (Goldstein et al., 2015), Individual Conditional Importance (ICI) (Casalicchio et al., 2019), Local Interpretable Model-agnostic Explanations (LIME) (Ribeiro et al., 2016), Shapley Additive Explanations (SHAP) (Shapley et al., 1953; Lundberg, 2017), Anchors (Ribeiro et al., 2018), and Counterfactual Expla-

nations (Wachter et al., 2017; Dandl et al., 2020). Global importance extensions often can be obtained by aggregating these local measures across all samples. SHAP and LIME are particularly popular because of their theoretical foundations, applications to a variety of data types, and their intuitive results (Molnar, 2020).

SHAP is a prediction-based local importance method that decomposes a model prediction into additive feature contributions. SHAP values extend the Shapley value framework from cooperative game theory, in which calculations are obtained by considering coalitions of predictors (Lundberg, 2017; Shapley et al., 1953). For a given observation, the contribution of a feature is evaluated by comparing model predictions across subsets of included and excluded variables, with the resulting marginal contributions averaged over all possible coalitions. Because SHAP integrates over varying configurations of the remaining predictors, it can capture aspects of both marginal and interaction structures. Modern implementations such as TreeSHAP and DeepSHAP are computationally efficient and yield precise approximations for tree-based and neural network models, respectively, contributing to the widespread adoption of SHAP in interpretable machine learning (Lundberg et al., 2018; Shrikumar et al., 2017).

LIME is another popular prediction-focused method, which approaches local explanation through interpretable surrogate models, such as linear regression or decision trees, to explain individual predictions from black-box models (Ribeiro et al., 2016). LIME generates a new dataset by perturbing predictor values around an observation of interest and obtaining the corresponding predictions from the black-box model (Molnar, 2020). An interpretable surrogate model is then fit to the perturbed data, with observations weighted according to their proximity to the target observation. Local feature importances are subsequently derived from the fitted surrogate model coefficients or splitting structure. Because a separate surrogate model is constructed for each observation, LIME produces individualized explanations while remaining model-agnostic and computationally flexible. Modern implementations additionally allow customization of perturbation strategies, kernel widths, discretization procedures, and feature selection mechanisms, all of which can substantially influence the resulting explanations. However, we show in our experiments that hyperparameter tuning is often required to detect localized interaction effects as the default LIME settings tend to emphasize marginal feature behavior (see Appendix A). Hyperparameter tuning for LIME is also difficult, as users typically do not have access to ground truth information on variable interactions.

ICI extends permutation-based global variable importance (PaP and Model-Reliance) to the local level by measuring how prediction error changes for an individual observation when a single feature is permuted (Casalicchio et al., 2019). ICI values are generally plotted on a curve that shows how the local importance varies across permutations, and its aggregate—Partial Importance (PI)—summarizes how feature relevance may change across the feature domain. ICI was designed to overcome the limitations of global permutation importance by revealing heterogeneous or conditional effects that may indicate interactions (Casalicchio et al., 2019). However, ICI requires the analyst to specify conditional groupings or interaction structures to meaningfully interpret different importance curves (Molnar et al., 2020). In practice, ICI can also exhibit high variance, requiring repeated permutations for every feature and observation, thus becoming computationally expensive for moderate to large datasets. We also note that ICI evaluates local importance by jointly permuting all predictors except $j$, while preserving variable $j$ and the response. This contrasts with classical permutation procedures, where variable $j$ is permuted directly and the dependence structure among the remaining predictors and the response is otherwise maintained. Our experiments show that this permutation structure struggles to reliably detect conditional importance when interacting features are not manually pre-identified.

Compared to other local importance techniques, SHAP, LIME, and ICI stand out because they provide an importance value for every observation in a dataset. For this reason, we focus primarily on comparing our results with these methods. Despite the popularity and flexibility of these methods, there remain many open problems in local explanation for further study and improvement. In particular, in our experiments, default SHAP and LIME, and uninformed ICI routinely assign a non-zero importance in regions of the feature space where the response is invariant to changes in a variable. While this is not inherently a flaw in the design of these methods, it may lead to inaccurate local interpretations in some applications. Additionally, extending prediction-based decomposition methods such as SHAP and LIME to multi-category classification settings introduces further complexity, since importance must be defined separately for each response class.

Collectively, these and the previously mentioned challenges highlight the need for a new complementary local importance technique that can reliably detect conditional effects without manual specification in the model loss setting, avoid spurious attributions, remain robust to hyperparameter tuning, and provide increased stability for importance metrics.

To address these limitations, we propose Conditional Local Importance by Quantile Expectations (CLIQUE), a new framework for assessing local variable importance. CLIQUE is specifically designed to assign zero importance to locally invariant features and detect interaction-driven effects without requiring user input. By grounding local importance in quantile expectation comparisons, CLIQUE offers a principled complementary perspective to existing local and global approaches, with particular advantages in stability and interpretability across regression, binary, and multi-class classification settings. CLIQUE is not intended to replace prediction-based explanation methods such as SHAP or LIME. Rather, it provides an alternative error-based approach that may be particularly useful for studying conditional or locally dependent feature importance.

Throughout this work, we use the term "conditional" in an interaction-based or context-dependent sense, rather than referring to conditional relationships in the formal probabilistic sense. Specifically, a variable is conditionally important when its effect depends on local regions of the predictor space due to the influence of other variables. The remainder of this paper defines CLIQUE, establishes its key properties, and demonstrates its empirical performance and novel contributions across a range of simulated and real-world scenarios.

## 2 Local Variable Importances with CLIQUE

### 2.1 CLIQUE Definition

Consider a dataset $(\mathbf{x}_1, y_1), \ldots, (\mathbf{x}_n, y_n) = (X, Y)$, where $X \in \mathbb{R}^{n \times p}$ is the data matrix and $Y \in \mathbb{R}^n$ contains the labels. Unlike classical permutation measures such as PaP and Model-Reliance, which evaluate global importance, CLIQUE focuses on local effects, yielding observation-level importance measures. Like ICI, CLIQUE is model-agnostic and evaluates observation-specific changes in error. However, CLIQUE differs fundamentally in that it employs a classical replace-and-assess technique in which the variable $j$ is permuted and all other variables remain fixed. Additionally, CLIQUE defines local importance in terms of cross-validated errors rather than changes in predictions from a single fitted model. CLIQUE also replaces random permutations with robust quantile-grid replacements (motivated by ICE curves of Goldstein et al. (2015)), improving stability and substantially reducing the number of perturbations required for reliable importance estimates (see Section 3.6).

Mathematically, consider the data point $\mathbf{x}_i$ with its corresponding CV model. We aim to compute the local importance for variable $j$, which we denote as $V_{ij}$. Let $\hat{f}(\mathbf{x}_i)$ represent the original prediction of $y_i$ given $\mathbf{x}_i$, obtained from the CV model trained on a fold that excludes $\mathbf{x}_i$. Let $\tilde{\mathbf{x}}_i(j, m)$ denote the altered version of $\mathbf{x}_i$, where the $j$th variable is replaced by the $m$th value from its quantile grid, constructed between the minimum and maximum of the $j$th variable in the training data. The CLIQUE value is

$$V_{ij} = \frac{1}{M} \sum_{m=1}^{M} \mathcal{L}\left(\hat{f}\left(\tilde{\mathbf{x}}_i(j, m)\right), y_i\right) - \mathcal{L}\left(\hat{f}(\mathbf{x}_i), y_i\right), \tag{1}$$

where $M$ is the number of altered values in the quantile grid and $\mathcal{L}$ is the loss function of interest. Quantile-based interventions provide a deterministic and distribution-driven approach for probing local sensitivity, while avoiding the instability introduced by random sampling or binning. Also, because CLIQUE defines local importance in terms of performance rather than prediction changes, it extends naturally to multi-class classification settings, where prediction-based local importance measures can be difficult to interpret. A complete workflow for CLIQUE is shown in Algorithm 1. Note that $V_{\cdot j}$ refers to the $j$th column in $V$.

We note that CLIQUE captures conditional information in the sense that an effect is conditional if it depends on, or changes with, another effect. Thus, we are not referring to conditional information in a formal probabilistic sense. We also note that for our work, "local" suggests at least some location-dependence on

---

**Algorithm 1** CLIQUE

---

1: **Input:** data $\underset{n \times p}{X}$, labels $\underset{n \times 1}{Y}$, model, quantiles $M$
2: $mod \leftarrow \text{model}(X, Y)$
3: $cvmod \leftarrow$ individual CV models
4: $\underset{n \times 1}{Err} \leftarrow \mathcal{L}(\text{predict}(cvmod, X), Y)$
5: $\underset{n \times p}{V} \leftarrow \underset{n \times p}{0}$
6: **for** $j = 1$ **to** $p$ **do**
7:     $\underset{M \times 1}{grid} \leftarrow M$ quantile values for variable $j$
8:     **for** $m = 1$ **to** $M$ **do**
9:        $\underset{n \times p}{W} \leftarrow \underset{n \times p}{X}$
10:       $\underset{n \times 1}{W_{:j}} \leftarrow grid[m]$
11:       $\underset{n \times 1}{\widetilde{Err}_j} \leftarrow \mathcal{L}(\text{predict}(cvmod, W), Y)$
12:       $\underset{n \times 1}{V_{:j}} \leftarrow V_{:j} + \frac{\widetilde{Err}_j - Err}{M}$
13:     **end for**
14: **end for**

---

the importance of a variable. If such location-dependence is absent, local importance should largely agree with global importance measures, which can be assessed using existing global methods.

## 2.2 CLIQUE Properties

Our CLIQUE algorithm is designed to satisfy several properties when assessing local importance:

**P1** If the model output for a given observation is invariant to changes in a variable, then CLIQUE assigns that variable zero population importance and negligible empirical importance for that observation.

**P2** CLIQUE importance values exhibit stability with relatively low variances.

**P3** CLIQUE is model-agnostic.

**P4** CLIQUE applies directly to multi-class problems without extensive modification.

**P5** CLIQUE values offer meaningful and intuitive aggregation across observations to assess subgroup/global behavior.

**P6** CLIQUE is computationally competitive with other local importance methods.

**P7** CLIQUE defines importance in terms of model error rather than predictions.

**P8** CLIQUE evaluates importance on non-training data.

These properties collectively outline the criteria that guide our formulation of a new local importance measure. **P1** expresses an intuitive notion of importance. While not strictly required for all local importance methods, this property carries strong conceptual merit, since assigning nonzero importance to variables with no influence can risk undermining interpretability in some cases (Fisher et al., 2019; Hooker et al., 2021). This is especially true when only a small subset of variables is truly important for a given datapoint. In such cases, unimportant variables may be incorrectly flagged as important. In our empirical results, CLIQUE more closely aligns with **P1**, while other methods' default settings often reflect additional prediction-based or marginal information that can yield nonzero local importance values even when local invariance is present. Furthermore, the following proposition and accompanying proof formally shows that CLIQUE assigns zero importance to locally invariant features and thus satisfies **P1**.

**Proposition 1** (Feature invariance). *Let $\hat{f}$ be a fixed prediction function and $\mathcal{L}$ be any loss function. If, for a given observation $\mathbf{x}_i$, the prediction $\hat{f}(\mathbf{x}_i)$ does not depend on feature $j$, then the corresponding CLIQUE importance $V_{ij} = 0$.*

*Proof.* By assumption, for all quantile-based replacements $\tilde{\mathbf{x}}_i(j, m)$ that modify only feature $j$ for $x_i$,

$$\hat{f}(\tilde{\mathbf{x}}_i(j, m)) = \hat{f}(\mathbf{x}_i). \tag{2}$$

Substituting this into the definition of CLIQUE (Equation 1) yields

$$\begin{aligned} V_{ij} &= \frac{1}{M} \sum_{m=1}^{M} \mathcal{L}\left(\hat{f}\left(\tilde{\mathbf{x}}_i(j, m)\right), y_i\right) - \mathcal{L}\left(\hat{f}(\mathbf{x}_i), y_i\right) \\ &= \frac{1}{M} \sum_{m=1}^{M} \mathcal{L}\left(\hat{f}(\mathbf{x}_i), y_i\right) - \mathcal{L}\left(\hat{f}(\mathbf{x}_i), y_i\right) \\ &= \mathcal{L}(\hat{f}(\mathbf{x}_i), y_i) - \mathcal{L}(\hat{f}(\mathbf{x}_i), y_i) = 0. \end{aligned} \tag{3}$$

$\square$

This proposition mathematically illustrates how CLIQUE's formulation ensures satisfaction of **P1**. Although the result holds exactly at the population level, an approximate analogue also holds in finite-sample or noisy settings provided the loss function is continuous, as is the case for most commonly used losses. In subsequent simulations, we show that this behavior is observed empirically for CLIQUE.

**P2** is inherently desirable, provided that it does not come at the cost of extra bias or excessive computation. It is motivated by the idea that the best importance results are achieved by replacing a specific value with all other possible variable distribution values. Unfortunately, this practice is quite computationally expensive (Molnar, 2020). Our use of quantile-replacements in CLIQUE, rather than permuting or dropping variables, offers an efficient proxy for comparing individual points to the entire variable distribution. To show that CLIQUE's quantile grid can outperform repeated permutations, we include simulations in Section 3.6 that offer comparisons between CLIQUE and a variation of CLIQUE that we refer to as Local Permute. Local Permute and CLIQUE are effectively identical, except that Local Permute performs $M$ variable permutations, while CLIQUE replaces a variable with $M$ quantile values.

**P3** is a standard desideratum in this regime of research, offering increased applicability for a method. To briefly illustrate the model-agnostic extensions of CLIQUE, we provide results in Section 3 for Random Forests, XGBoost, and Artificial Neural Network models.

**P4** is advantageous because it allows CLIQUE to extend naturally to multi-class settings, avoiding the need for specialized derivations or one-vs-all decompositions as is common for methods such as LIME and SHAP. This makes CLIQUE broadly applicable and computationally efficient, while providing a unified treatment of variable importance across all predicted classes. To demonstrate the application of CLIQUE to multi-class settings, we provide analyses based on the MNIST classification task in Section 4.3 and Appendix E.

**P5** is a valuable property for local importance methods because it connects observation-level explanations to coherent subgroup or global summaries. The creators of SHAP emphasize that consistent local attributions should naturally aggregate to form reliable global importance measures (Lundberg, 2017). Similarly, CLIQUE offers very natural aggregation across subgroups through classical summary metrics such as means, medians, and measures of dispersion. In this way, CLIQUE satisfies **P5**, allowing it to provide both fine-grained local diagnostics and broader group assessments. We illustrate several applications for this in Section 4.

**P6** is inherently desirable, provided this efficiency does not come at the cost of additional bias or variance. To illustrate that CLIQUE satisfies **P6**, we include a simulation in Appendix F that offers a time comparison between CLIQUE, ICI, SHAP, and LIME.

**P7** is not inherently superior or inferior to importances that are interpreted on predictions. Rather, it highlights the type of information CLIQUE is designed to capture, distinguishing it from prediction-based approaches like LIME and SHAP. Error-based importance can be particularly valuable for diagnosing model weaknesses and identifying features that most contribute to local misfit (Rudin, 2019). This perspective offers practitioners an alternative way to examine model behavior, complementing existing prediction-focused interpretability tools.

Finally, **P8** aligns with best practices in interpretable machine learning and model evaluation, ensuring that importance is assessed on data outside of the fitting process (Molnar, 2020). This reduces overfitting concerns and helps importance reflect generalizable behavior rather than artifacts of the training set. To capitalize on the benefits associated with **P8**, CLIQUE computes local importance using CV errors by training multiple CV models with the same hyperparameters determined for the full model. Each variable is systematically modified for every data point and predictions are generated from a CV model that excluded that point from training. Comparing the CV error rate of the modified data to the original CV error rate yields the local importance measure.

## 3 Simulated Experiments

To demonstrate how CLIQUE compares with LIME, SHAP and ICI in capturing locally dependent relationships, we apply the methods to simulated data with known structures. For all analyses, we use $M = 25$ quantiles for CLIQUE and 25 permutations for ICI. This value was selected to provide additional margin beyond the stability behavior observed for $M = 10 - 20$ in Section 3.6, while also leveraging variance reduction associated with larger sample averages under the Central Limit Theorem. Further exploratory analysis of the impact of $M$ on CLIQUE values and their variance is provided in Section 3.6. Additionally, unless otherwise specified we use a standard randomly assigned 5-fold CV architecture. We use default parameters for SHAP and increase the number of bins for LIME to 10. For each of the following three simulations, all features are independent and identically distributed (iid) from a uniform distribution $\mathcal{U}(-1, 1)$, and we generate 400 training observations.

Though CLIQUE generalizes to a number of different loss functions, in our implementation of Algorithm 1, we use absolute error for its robustness to extreme values (See Appendix B for comparison to squared loss). The prediction function returns class probabilities rather than raw labels for all models and datasets in Sections 3 and 4, except for the neural network examples in Section 3.2 and 3.4. When raw labels are used, absolute error reduces to a 0–1 loss. When probabilities are used, the response is one-hot encoded, and absolute error is computed for each class probability and averaged across classes in a modified Brier Score.

### 3.1 AND Gate Data

We begin with a simple simulation based on an AND gate (Figure 1). To create this data, we generate three features and produce the response using:

$$y = \begin{cases} 1, & \text{if } v_1 > -\frac{1}{3} \ \& \ v_2 > -\frac{1}{3} \\ 0, & \text{otherwise.} \end{cases} \tag{4}$$

Note that $v_3$ has no influence on the output.

We build a Random Forest model using the randomForest package (Liaw & Wiener, 2002) in the R programming language (R Core Team, 2022). From this model, we extract ICI values using the `featureImportance` package (Casalicchio, 2025), LIME values using the `lime` package (Hvitfeldt et al., 2022), and SHAP values using the `treeshap` package (Komisarczyk et al., 2024).

Based on the geometry of the data in Figure 1, when $v_2 < -1/3$ the response is effectively insensitive to changes in $v_1$. Thus, the CLIQUE importance of $v_1$ should be near zero in this region. In contrast, when $v_2 > -1/3$, variation in $v_1$ has a substantial effect on the model output, and its importance should therefore be strongly positive. From Figure 1, we see this in the CLIQUE values. When $v_2 < -1/3$, the importance of $v_1$ is practically zero, while for $v_2 > -1/3$, $v_1$ has a positive importance. This information stands in contrast

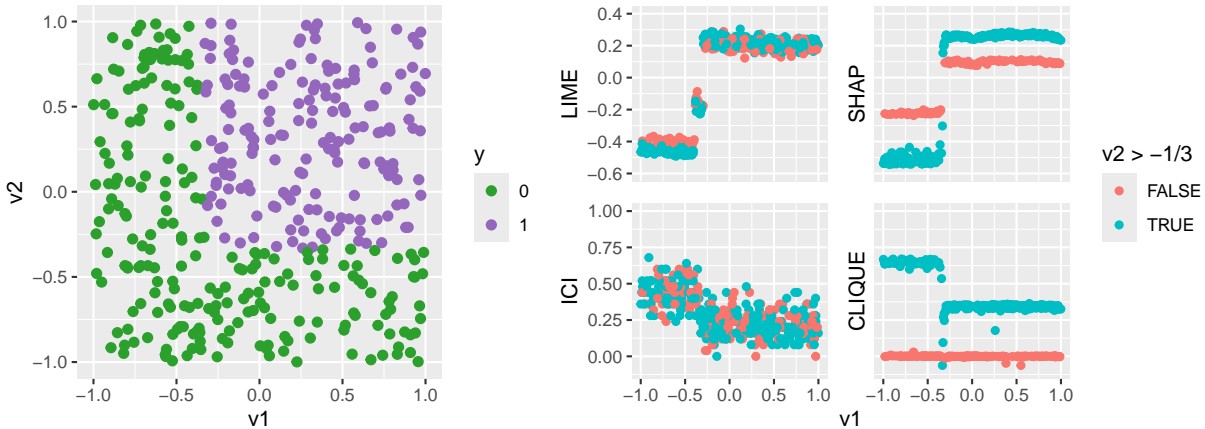

Figure 1: Scatterplot results for the AND gate data. Left-half: Interaction predictors $v_1$ and $v_2$ colored by the label $y$ as computed in Equation 4. When $v_2 < -1/3$, $y$ is invariant to any change in $v_1$. Right-half: $v_1$ Local variable importances colored by whether $v_2 > -1/3$. CLIQUE values output zero importance when $v_2 < -1/3$ and a nonzero importance otherwise. LIME, SHAP, and ICI each yield non-zero assessments of $v_1$ when $v_2 < -1/3$. Additionally, ICI and this implementation of LIME offer a much more global perspective here.

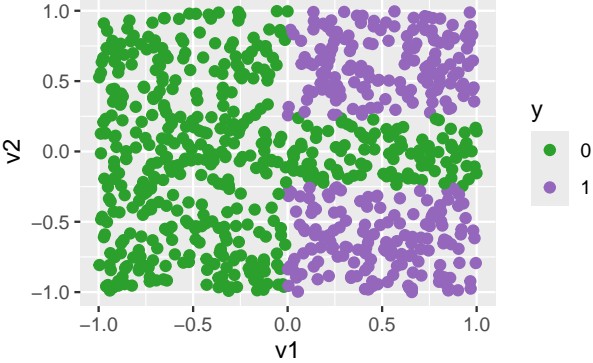

Figure 2: Scatterplot of the Corners data colored by the label $y$ (see Equation 5). $v_2$ is conditionally unimportant when $v_1 < 0$. $v_1$ is conditionally unimportant when $|v_2| < 1/4$.

to LIME and ICI which show no discernible differences in the importance of $v_1$ relative to $v_2$, indicating that while they do capture the global structure well, they do not capture the same conditional information in the data as CLIQUE. We note that LIME can capture local structure well with much smaller kernel widths (see Appendix A), but default hyper-parameters tend to struggle. SHAP falls between LIME and CLIQUE. It displays two distinct trends for $v_1$, but still assigns nonzero importance when $v_2 < -1/3$. In other words, CLIQUE aligns closely with **P1**, whereas LIME, ICI, and SHAP continue to attribute importance that reflects broader marginal information.

## 3.2 Corners Data

The AND Gate is a symmetric problem, where the results for $v_1$ mirror the results for $v_2$. We now simulate values for a non-symmetric dataset that we call the Corners data (Figure 2). We generate three features

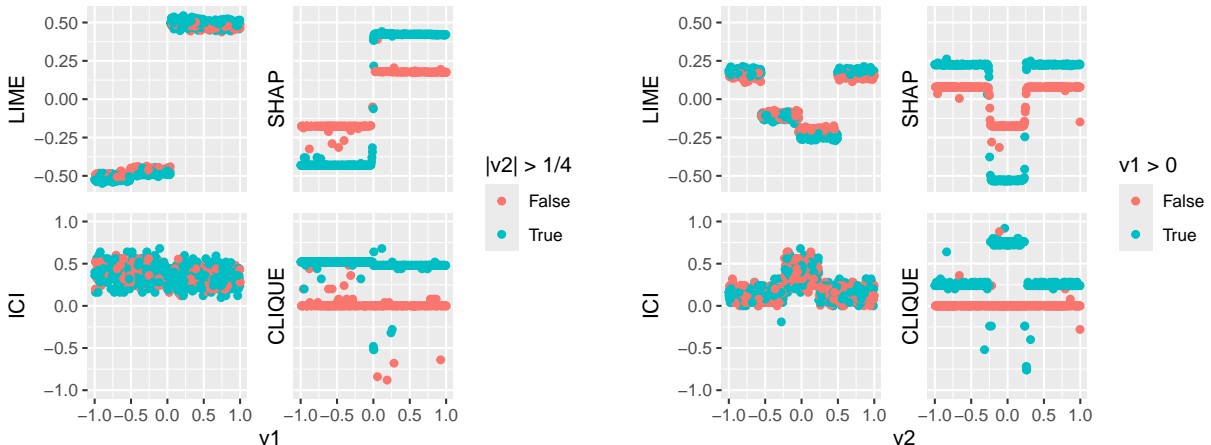

Figure 3: Scatterplots of local variable importances vs. variable values from the Corners data experiment. Left-half: $v_1$ importances colored by whether $|v_2| > 1/4$. Right-half: $v_2$ importances colored by whether $v_1 > 0$. As in Figure 1, the CLIQUE values generally emphasize the intended conditional relationships between $v_1$ and $v_2$, while LIME, ICI, and SHAP do not offer this same information with respect to **P1**.

with response:

$$y = \begin{cases} 1, & \text{if } v_1 > 0 \ \& \ |v_2| > \frac{1}{4} \\ 0, & \text{otherwise.} \end{cases} \tag{5}$$

In this data, we expect that for CLIQUE, $v_2$ is not important when $v_1 < 0$ and $v_1$ is not important when $|v_2| < 1/4$. To demonstrate **P3**, the model-agnostic capability of CLIQUE, we develop a basic Artificial Neural Network (ANN) structure in Python for modeling the Corners data and extract LIME, SHAP (via deepSHAP), ICI, and CLIQUE values from five randomly partitioned CV folds, which are shown in Figure 3.

The behaviors of each of the importance methods match the trends found in Figure 1. The LIME values again show marginal prediction information, while the ICI values show marginal importance. SHAP values offer two distinct conditional trends, but assign nonzero impact when $|v_2| < 1/4$ and $v_1 < 0$. CLIQUE provides conditional importance values, with effectively zero importance when $|v_2| < 1/4$ and $v_1 < 0$. Additionally, it gives a distinctly positive importance for $v_1$ when $|v_2| > 1/4$ and for $v_2$ when $v_1 > 0$. That is, CLIQUE emphasizes the conditional structure embedded in this simulation **P1**, whereas the other methods place greater emphasis on marginal behavior.

### 3.3 Regression Interaction Data

We now turn to a location-dependent regression simulation (Figure 4) with four features that produce the response:

$$y = \begin{cases} v_1 + \epsilon, & \text{if } v_3 > 0 \\ v_2 + \epsilon, & \text{if } v_3 < 0. \end{cases} \tag{6}$$

where $\epsilon \sim \mathcal{N}(\text{mean} = 0, \text{sd} = 0.01)$.

In this data, $v_1$ is conditionally unimportant when $v_3 < 0$ and $v_2$ is conditionally unimportant when $v_3 > 0$. We build a Random Forest model and again extract LIME, SHAP, ICI and CLIQUE values, with the results shown in Figure 4. The trends are similar to previous findings: CLIQUE offers information reflecting property **P1** which is novel with respect to information from LIME, SHAP, and ICI. As in Section 3.1, SHAP values show differences between the two regions, but most values remain nonzero for the region where no $v_1$ importance is expected.

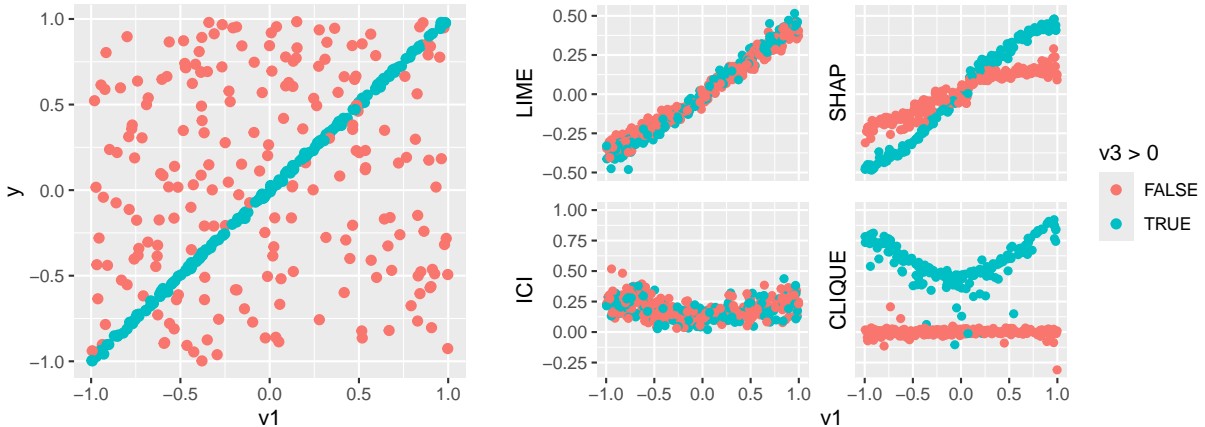

Figure 4: Scatterplots of the response variable (left-half) and local variable importances (right-half) vs. $v_1$ for the Regression Interaction data. Each plot is colored by whether $v_3 > 0$ (see Equation 6). $v_1$ is conditionally unimportant when $v_3 < 0$. $v_2$ is conditionally unimportant when $v_3 > 0$. CLIQUE more strongly emphasizes these known conditional relationships between $v1$ and $v3$, while LIME, SHAP, and ICI reflect mixtures of global, marginal, and conditional structures.

## 3.4 Higher Dimensional Correlated Data

Next we extend our conditional regression to a substantially higher-dimensional and more complex setting with 1000 observations and 101 features. To do this we simulate 100 variables from a multivariate normal distribution with mean 0 and variance 1. We also impose a common correlation between all pairs of variables of $\rho = 0.5$. Finally, we simulate a single location-driving variable $z$ from a binary draw with probability of 0.5. From these we create the response:

$$y = \begin{cases} 3(v_1 + v_2 + v_3 + v_4 + v_5) + \epsilon, & \text{if } z = 1 \\ 3(v_6 + v_7 + v_8 + v_9 + v_{10}) + \epsilon, & \text{if } z = 0, \end{cases} \tag{7}$$

where $\epsilon \sim \mathcal{N}(\text{mean} = 0, \text{sd} = 0.1)$.

In this setting, all variables possess some predictive information due to the imposed correlation structure. Specifically, 10 variables exhibit direct conditional signals, while the remaining variables inherit weaker predictive associations through correlation with the active predictors. However, the imposed structure implies that $v_1 - v_5$ are conditionally more important when $z = 1$, whereas $v_6 - v_{10}$ are conditionally more important when $z = 0$. Importantly, $z$ determines which subset of variables drives the response rather than contributing a direct marginal effect itself.

Because of this and the high dimensionality of the problem, we fit an ANN with three randomly partitioned CV folds to extract local importance metrics for $v_1$ and $v_6$ (similar results are observed when considering other variables), with results shown in Figure 5. For both $v_1$ and $v_6$, CLIQUE and LIME exhibit noticeable differences across values of $z$. However, the default implementation of LIME assigns larger importance values to both variables when $z = 1$, while SHAP and ICI appear to have negligible differences. In contrast, CLIQUE appears to more closely reflect the imposed conditional structure, indicating that $v_1$ is more important when $z = 1$, whereas $v_6$ is more important when $z = 0$.

Additional simulations examining alternative correlation structures ($\rho \in \{0, 0.25, 0.5\}$) together with settings in which $z$ also exhibits a direct marginal effect are provided in Appendix C. These supplementary results show qualitatively similar behavior across the examined configurations.

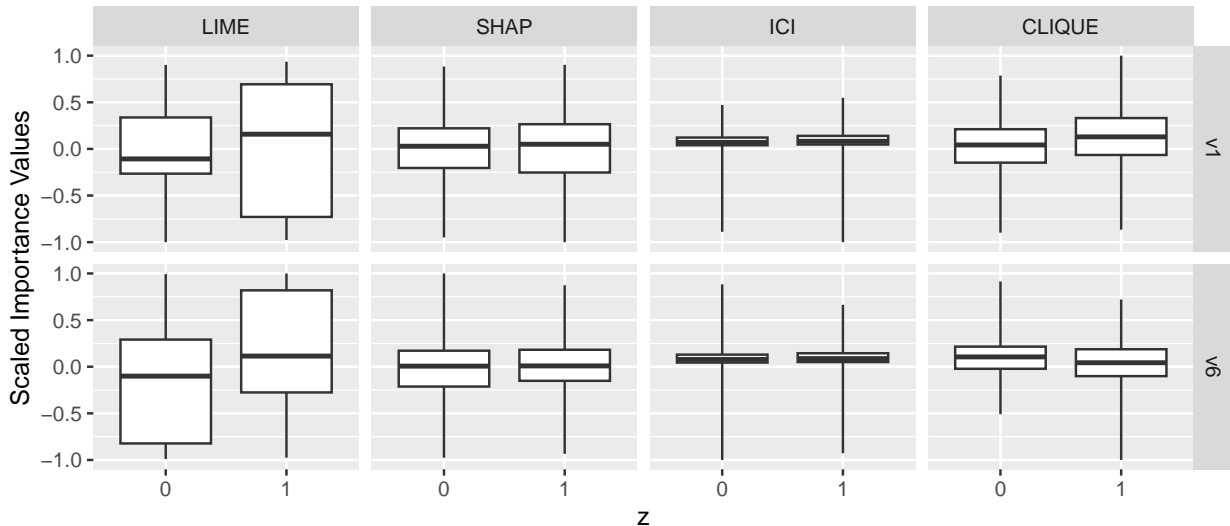

Figure 5: Distributions for $v_1$ and $v_6$ importances across $z$. Importances are scaled by their maximum absolute value within each method. Classically defined outliers are included in whiskers. The CLIQUE values best align with the imposed conditional structure of Equation 7.

## 3.5 Quantifying Results

While Figures 1, 3, and 4 illustrate qualitative differences in local importance behavior under known data-generating mechanisms, we additionally quantify the extent to which these methods assign importance to features with no conditional effect on the response. To this end, we compute the mean absolute error (MAE) with respect to **P1**, defined as the mean absolute difference between estimated local importance values and zero conditional importance, after standardization ($\frac{v}{|\max(v)|}$) to ensure comparability across methods and subsetting to only observations where $y$ is invariant to a change in $v1$.

|  | LIME | SHAP | ICI | CLIQUE |
|---|---|---|---|---|
| AND Gate | $0.411 \pm 0.010$ | $0.232 \pm 0.006$ | $0.418 \pm 0.011$ | $0.009 \pm 0.001$ |
| Corners-v1 | $0.867 \pm 0.006$ | $0.358 \pm 0.009$ | $0.528 \pm 0.010$ | $0.037 \pm 0.004$ |
| Corners-v2 | $0.599 \pm 0.022$ | $0.164 \pm 0.006$ | $0.241 \pm 0.011$ | $0.010 \pm 0.001$ |
| Corners-Regression | $0.280 \pm 0.005$ | $0.168 \pm 0.004$ | $0.388 \pm 0.008$ | $0.043 \pm 0.002$ |

Table 1: Comparison of mean absolute errors (MAE) from an importance of zero, for each dataset computed across 50 Monte-Carlo simulations. 95% confidence intervals are also provided based on the repeated simulations. Errors are computed, following standardization, between true zero importance values and estimated local importance values for each method.

Table 1 summarizes MAE from zero across all simulation settings. This comparison is specific to the **P1** criterion and should not be interpreted as a general ranking of local importance methods. In every scenario, CLIQUE exhibits substantially smaller errors with respect to **P1** than competing methods, often by an order of magnitude. These results empirically reinforce **P1**, demonstrating that CLIQUE not only satisfies feature invariance in theory, but also generally achieves it in practice—an important consideration for interpretability and trust in local explanation methods.

## 3.6 Effect of Parameters on Permutation vs. Quantile Replacement

To assess the advantages of quantile-based importance, we compare Local Permute (CLIQUE with permutation instead of quantile replacement) to CLIQUE. We examine the effect of $M$ (the number of replacements)

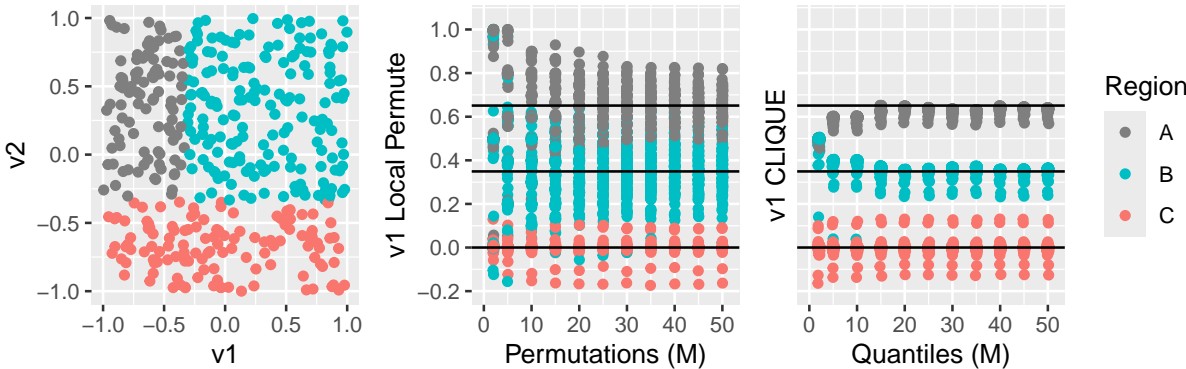

Figure 6: Plot showing data scatterplot regions and the relationship between importance value distributions and the hyper-parameter $M$. All plots are colored by regions of interest with respect to $V1$ and its importance values. "A" and "B" are regions where $V1$ should be important, but with different levels due to distribution. "C" denotes points where $V1$ should not be important. Left: Scatter-plot showing $V1$, $V2$, and their associated regions. Center: Plot showing the distribution of Local Permute values across the hyper-parameter $M$, colored by region. Right: Plot showing the distribution of CLIQUE values across the hyper-parameter $M$, colored by region. CLIQUE values achieve a distinctly smaller variance for each value of $M$.

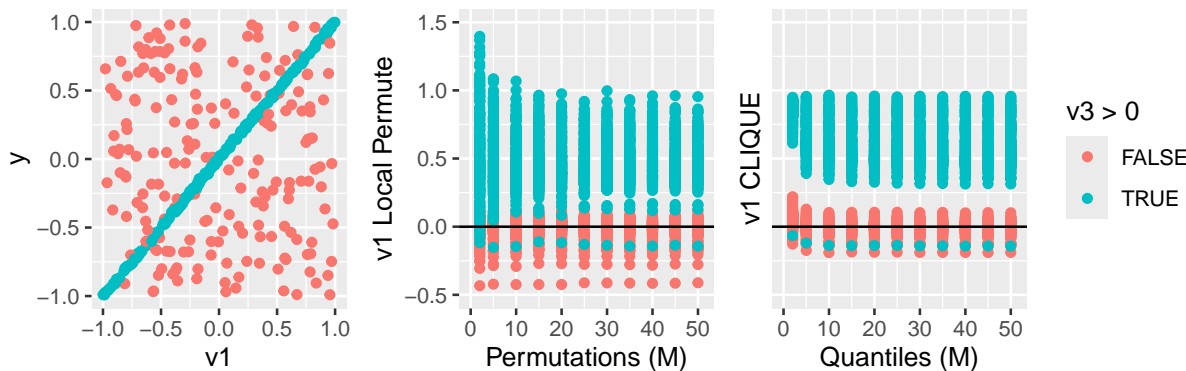

Figure 7: Plot showing data scatterplot regions and the relationship between importance value distributions and the hyper-parameter $M$. Plots are colored by regions of interest with respect to $V1$ and its importance values. The blue region represents where $V3 > 0$ and $V1$ should be important, while the red region denotes points where $V3 < 0$ and $V1$ have no effect on the response. Left: Scatter-plot showing $V1$, and $y$, colored by their associated regions. Center: Plot showing the distribution of Local Permute values across the hyper-parameter $M$, colored by region. Right: Plot showing the distribution of CLIQUE values across the hyper-parameter $M$, colored by region. CLIQUE values achieve a distinctly smaller variance for each value of $M$.

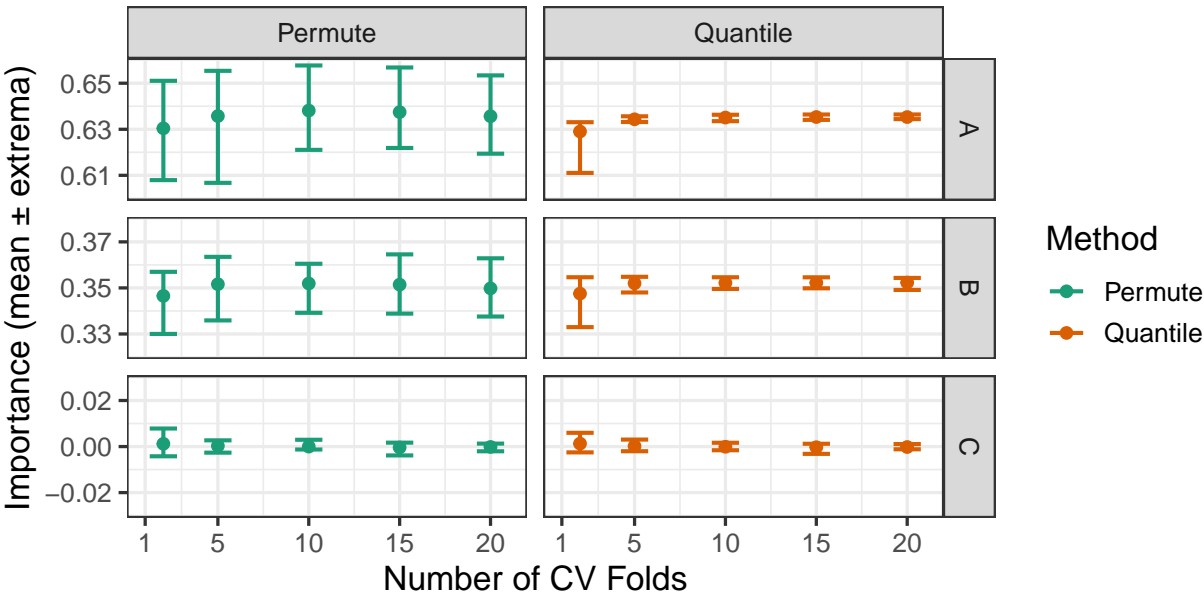

Figure 8: Plot showing means and extrema for each of the AND gate regions (see Figure 6) across both permutation and quantile-replacement architectures. All plots show strong stability for folds above 5. CLIQUE values achieve a distinctly smaller variance for two of the regions and for $K > 5$.

on these values for both the AND gate data from Section 3.1 and the Regression interaction data from Section 3.3. We then explore the impact of the number of CV-folds ($K$) on these same methods for the data regions of the AND gate data.

Figures 6 and 7 illustrate that CLIQUE achieves substantial gains with respect to **P2**, as Local Permute values exhibit much higher variance than CLIQUE values in all regions where $v_1$ is expected to have non-zero local importance. The results in the figures also provide guidance on selecting an appropriate $M$. As noted in Section 3, we chose $M = 25$ with the intent to stabilize estimates and reduce variance. Figure 6 and 7 reveal that CLIQUE variances remain remarkably consistent across all $M$, while the value centers show some initial bias and volatility before correctly stabilizing around $M$ of 15 or 20. Beyond this point, higher $M$ values provide diminishing returns at an increased computational cost.

Considering these findings alongside expected stability properties of the Central Limit Theorem, we suggest that $M$ values above 25 should offer sufficient margin beyond anticipated stability thresholds and are generally conservative enough to yield desirable estimates. However, larger $M$ values may be warranted for datasets with highly complex interactions or decision boundaries.

We now evaluate the impact of $K$ for the AND gate data across its respective regions and both Local Permute and CLIQUE. Figure 8 suggests that both methods exhibit strong stability for values at or above $K = 5$. This behavior indicates that extremely large fold counts may yield diminishing returns in this setting. Across the AND-gate regions, CLIQUE also exhibits reduced variability compared to Local Permute in regions $A$ and $B$ for $K > 5$, which aligns with results from Figure 6.

## 4 Real Data Experiments

Here we apply CLIQUE to real data, including the concrete regression dataset (Yeh, 1998), the lichen dataset (Cutler et al., 2007) and the MNIST dataset (Alpaydin & Kaynak, 1998). For all analyses, we again set $M = 25$ replacements for both CLIQUE and ICI, and again use a standard randomly assigned 5-fold CV architecture.

## 4.1 Concrete Regression

To illustrate an application of our method in a regression setting, we use the Concrete Compressive Strength data (Yeh, 1998). This dataset is available on the UCI data repository and consists of 1030 observations. There are eight quantitative ingredients or features and a response listed with their respective units as: Cement (kg/$m^3$), Blast Furnace Slag (kg/$m^3$), Fly Ash (kg/$m^3$), Water (kg/$m^3$), Superplasticizer (kg/$m^3$), Coarse Aggregate (kg/$m^3$), Fine Aggregate (kg/$m^3$), Age (days), and Concrete Compression Strength (MPa).

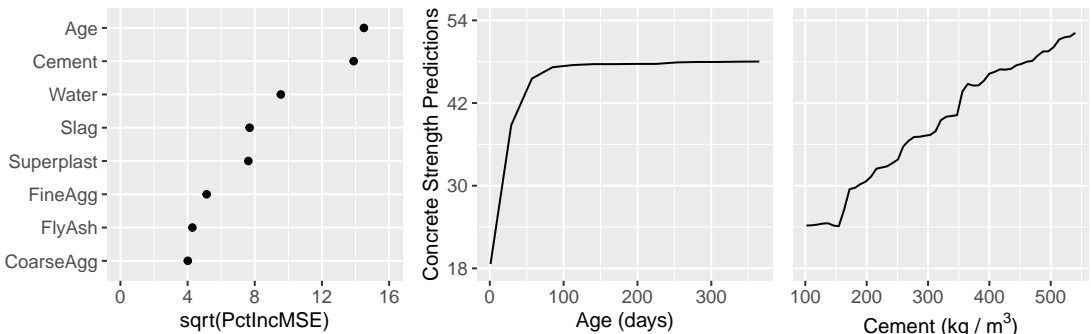

Figure 9: Left: Permutation-based global importances from a random forest model for all features in the Concrete data. Center-Right: PDPs for Age and Cement, the top two features.

With this data, we build a random forest model to predict Concrete Strength and extract LIME, SHAP, ICI, and CLIQUE values from it. Results for permutation-based global variable importances and PDPs are shown in Figure 9. These results show that Age and Cement appear to be the most important ingredients for determining Concrete Strength. We especially notice that the Cement PDP shows a consistent trend upward, whereas the Age PDP shows Strength predictions initially rising rapidly, but then stabilizing at an Age between 50 and 100 days. These results raise the question: Does Cement importance change for different Ages?

To answer this question, we divide our local importances into two groups based on whether the Age exceeded 75 days or not. The CLIQUE boxplots in Figure 10 show that Cement has a greater importance when Age is lower (**P5**). In other words, concrete strength accuracy is more influenced by cement amount in lower ages. Little difference is observed in the LIME or ICI values, while the SHAP values in Figure 10 do show a slightly larger spread when Age is less than 75. This effect is distinctly diminished compared to the difference seen

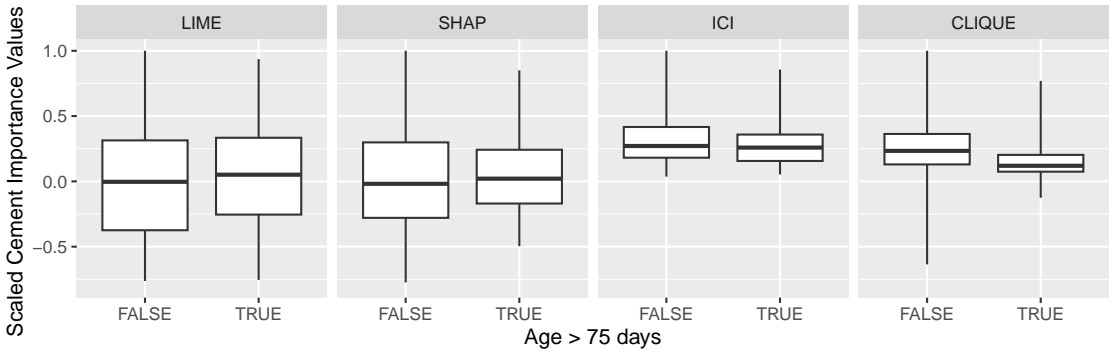

Figure 10: Distributions for Cement importances for different Age thresholds. Importances are scaled by their maximum absolute value within each method. Classically defined outliers are included in whiskers. The CLIQUE values exhibit a stronger separation between the two regions, reflecting the conditional importance perspective emphasized by CLIQUE.

in the CLIQUE distributions. Aggregate SHAP values, calculated by averaging the SHAP magnitudes, yield a ratio of 1.36, whereas the mean of the CLIQUE values produces a ratio of 1.68 times higher for lower ages compared to higher ages.

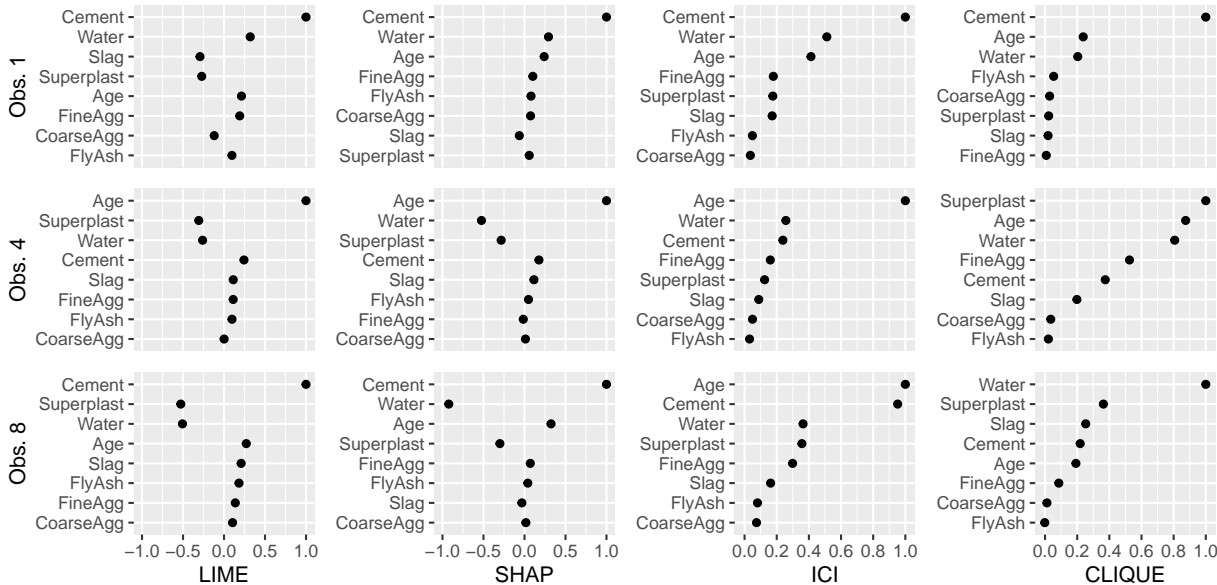

Figure 11: CLIQUE, SHAP, ICI, and LIME values of all predictor variables for three instances in the Concrete data. Importances are scaled by their maximum absolute value within each method. Based on the CLIQUE values, each of the selected observations has a distinctly unique set of important variables, which deviates substantially from the global trends.

This dataset provides an excellent opportunity to examine observation-level importance metrics. In Figure 11, we show LIME, SHAP, ICI, and CLIQUE values for three observations: the first, fourth, and eighth instances of the concrete dataset. For Observation 1, Cement definitively has the largest importance. A brief empirical check shows that the Cement importance for Observation 1 is the largest CLIQUE value across the entire dataset. The LIME and SHAP values for Cement are also large, which, when considered alongside the Cement PDP in Figure 9, suggests that the Cement value for this observation is high. In contrast, the other visualized observations do not have Cement as a top contributing variable according to CLIQUE. For Observation 4, CLIQUE indicates that Superplasticizer, Age, and Water are the most important features for determining Strength. The other importance methods generally agree, although they give higher priority to Age due to its strong global influence. Meanwhile for Observation 8, CLIQUE shows that Water is distinctly the most important, while the other methods again emphasize Age or Cement for their strong marginal effects.

## 4.2   Lichen Classification

In the lichen data, each observation is a different location in a survey region of the Pacific Northwest. This dataset consists of many climate, vegetation, and topographic features for assessing a binary response of the presence or absence of a species of lichen (Lobaria oregana). To further demonstrate **P3**, we build an XGBoost model using all available features, and extract LIME, SHAP, ICI, and CLIQUE values. Results for permutation-based global variable importances (Fisher et al., 2019) and selected PDPs are shown in Figure 12. MinTempAve (minimum temperature) and ACONIF (age of conifers) are the most important variables globally for classifying the presence of Lichen. The ACONIF PDP shows a clear upward trend, whereas MinTempAve exhibits a plateau, followed by a steep increase, and another plateau. We aim to determine whether the importance of ACONIF changes for different minimum temperatures. We use XG-

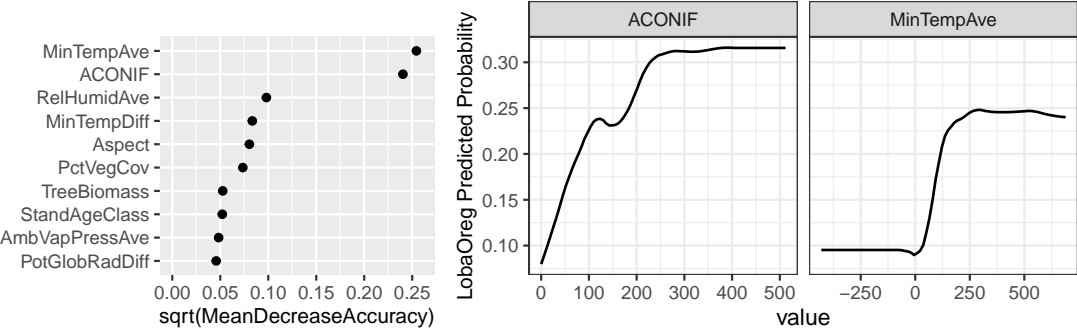

Figure 12: Global importances (left) and PDPs (right) for top XGBoost features in the Lichen data.

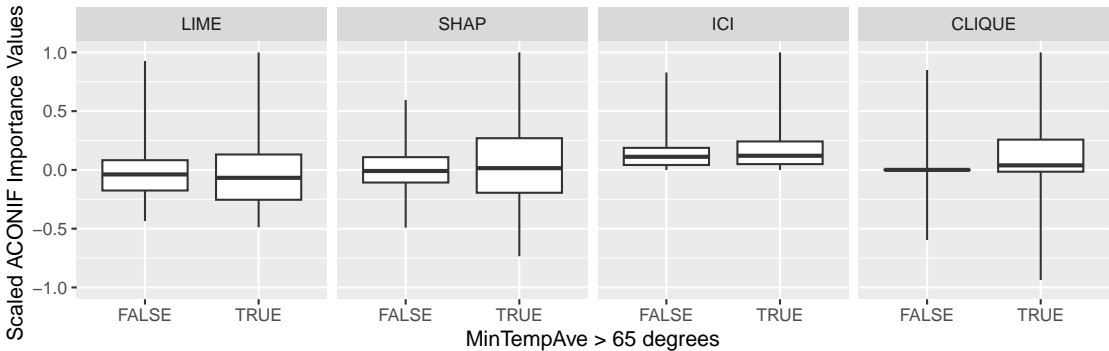

Figure 13: Distributions of XGBoost ACONIF importances for different MinTempAve thresholds. Importances are scaled by their maximum absolute value within each method. Classically defined outliers are included in whiskers. CLIQUE exhibits the strongest separation between these regions under this aggregation, suggesting that the method highlights this conditional relationship more prominently.

Boost to illustrate another model example of CLIQUE satisfying **P3** and can also offer a direct comparison between an XGBoost model and a Random Forest model (see Appendix D).

To do this, we split our data based on whether MinTempAve is below 65 and aggregate importances (**P5**). The resulting boxplots in Figure 13 show that ACONIF has a majority of CLIQUE importance values at essentially zero when minimum temperatures are lower. This makes sense, as lichen struggle to survive in areas with low or freezing temperatures, making ACONIF less important in those regions. If it is sufficiently warm for lichen to survive, then ACONIF more strongly influences their success. The SHAP value distributions in Figure 13 show apparent differences in spread when MinTempAve is above or below 65, but assign a greater aggregated importance to ACONIF for temperatures below 65 than the CLIQUE comparisons. In contrast, the LIME and ICI ACONIF importance boxplots are very similar across low and high MinTempAve conditions.

### 4.3 MNIST Digit Classification

As another illustrative example, we analyze a down-sampled version of the MNIST Digit test dataset (Alpaydin & Kaynak, 1998) available from scikit-learn (Pedregosa et al., 2011). This dataset consists of 64 features, each a pixel value, with x and y-coordinates ranging from 1 to 8. These pixel values help determine which hand-drawn digit each observation corresponds to. Since this is a multi-class problem, it offers a prime example for illustrating **P4** in CLIQUE, while SHAP and LIME are not natively adapted to provide local

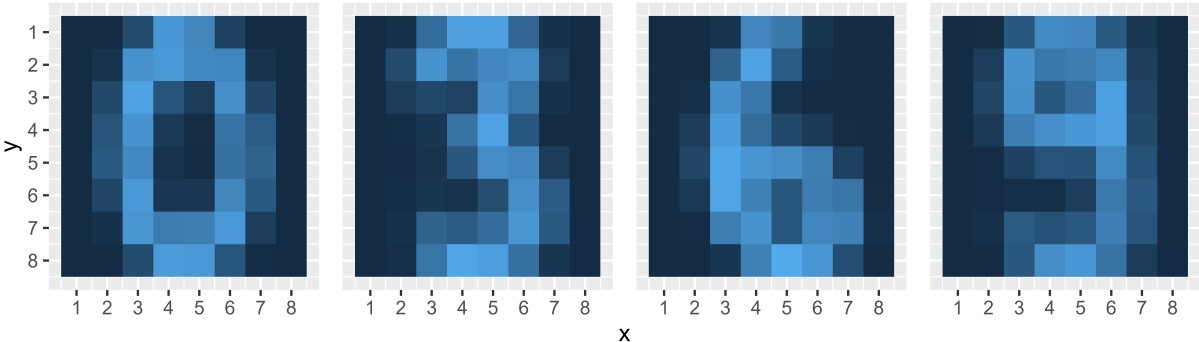

Figure 14: Aggregate tile plots for four digits from the down-sampled MNIST Digit test data. Pixel values are averaged across each observation of a particular digit class to generate these plots.

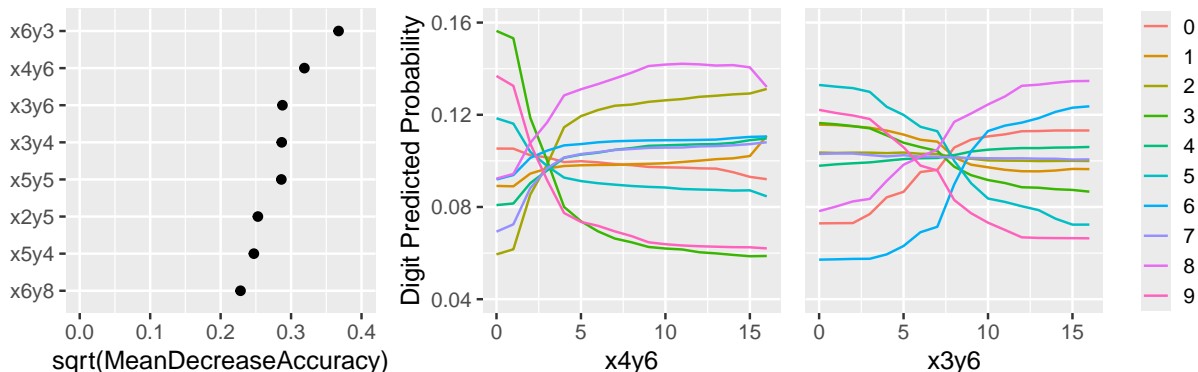

Figure 15: Global permutation importances (left) and Partial Dependence Plots (center & right) for top features in the downsampled MNIST Digits test data.

importances. An example plot of digits is shown in Figure 14. We train a Random Forest model on this data to obtain CLIQUE values. Results for the top permutation-based global importance variables and PDPs are shown in Figure 15.

The global results show that `x4y6` and `x3y6` are two of the most important variables for classifying digits and they neighbor each other. The `x4y6` PDP in Figure 15 shows a distinct region of changing probabilities for low values followed by a plateau effect after about 4.5. The `x3y6` PDP shows a clear crossing pattern at a value of about 7.5. These results suggest the possibility that `x4y6` may be more or less important for low or high values of `x3y6`. They especially imply that `x3y6` may be more or less important for low or high values of `x4y6`.

The left-panel boxplots in Figure 16 show that `x4y6` has a greater importance when `x3y6` is less than 7.5, while `x3y6` has a greater importance when `x4y6` is less than 4.5. These results imply an interaction between these pixels, where each pixel value is more important when the other has a lower value. This behavior is similar to results seen in the AND Gate data of Section 3.1.

In addition to comparing pixel importances against pixel values, we also compare them across digit labels. We present this from two different perspectives. The first is found in the right-panels of Figure 16 which shows CLIQUE boxplots grouped by labels, effectively illustrating that CLIQUE provides **P4** and **P5**. Both `x4y6` and `x3y6` vary in their importance across different digit labels. Pixel `x4y6` has its highest CLIQUE

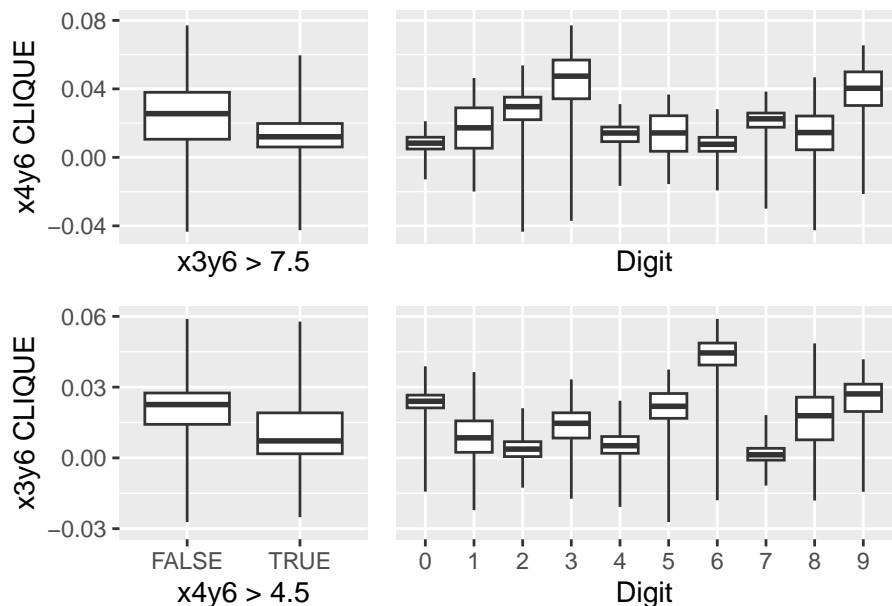

Figure 16: Distributions for x4y6 (top) and x3y6 (bottom) CLIQUE values across digit classes (left) and identified cutoffs (right). Classically defined outliers are included in whiskers, rather than as distinct points. Digit-aggregated plots show different levels of importance across class-levels. Cutoff-aggregated plots imply an interaction between these pixels, where each pixel value is more important when the other has a lower value.

values for the digits 3 and 9. Although initially surprising, further analysis reveals that for digits 3 and 9, x4y6 equals 0 more than 80% of the time, while for other digits, it equals 0 less than 60% of the time. Therefore, a value of 0 strongly indicates that a digit is likely a 3 or 9, whereas a value above 0 suggests otherwise. The results also show that x3y6 has particularly high CLIQUE values for the digit 6. These results suggest that x3y6 contains information useful for potentially distinguishing the digit 6 from all other digits. When considered alongside trends from Figure 15, low values of x3y6 appear to correspond with a distinctly lower likelihood that a digit is a 6 rather than another label.

We can do further analysis using a dimensionality reduction method known as PHATE (Moon et al., 2019). PHATE is a visualization tool that balances global and local structures in data. Here, we reduce the data down to two dimensions and create scatterplots of those dimensions in Figure 17. We color this plot by class labels and discretized CLIQUE categories to identify further trends and patterns in our data. From Figure 17, PHATE appears to separate the data into 12 natural clusters, which mostly align with the digit labels, with class 1 and 9 being split.

Beginning with x4y6 in Figure 17, we can see that high CLIQUE values tend to especially appear in the bottom-left quadrant of the plot. Low CLIQUE values tend to fit into the top-right section. Generally, digits 3 and 9 show high values, while digits 0 and 6 show low. One digit class especially catches our attention though: the number 5. Its cluster shows up as a gradient with low CLIQUE values in the top section and high CLIQUE values in the bottom. This suggests that x4y6 CLIQUE values may be a good metric for distinguishing between different ways of writing the number 5.

Surprisingly, x3y6 highlights opposing regions in Figure 17 despite its proximity to x4y6. High CLIQUE values generally appear in the bottom right quadrant, while low values fall in the top left. We can visually see that digits 2, 4, and 7 show low values, while digits 5, 6, and 9 show high values. The digits 3 and 9 have overlapping groups, yet we can fairly distinguish them by whether their x3y6 CLIQUE value was high or low.

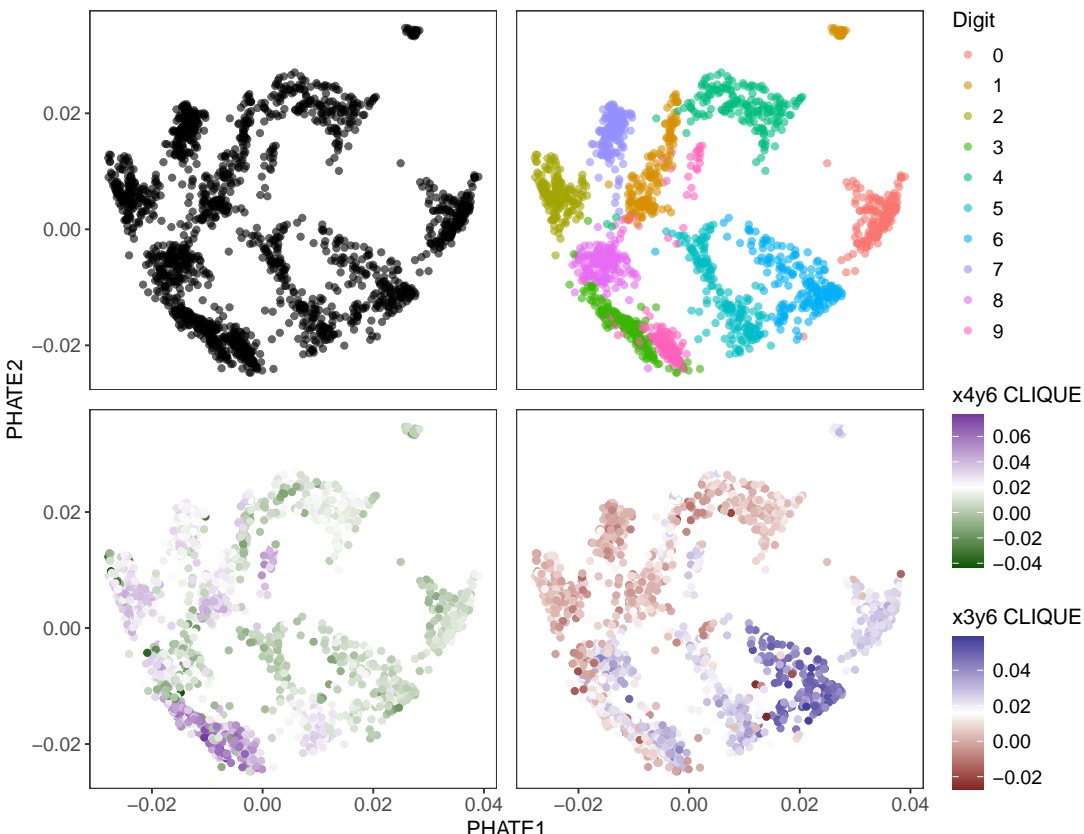

Figure 17: Scatterplots of PHATE embeddings for the MNIST Digit data. Top-Left: Default plot. Top-Right: PHATE colored by actual digit classes. Bottom-Left: PHATE colored by `x4y6` importances cut into three equally sized bins. Bottom-Right: PHATE colored by `x3y6` importances cut into three equally sized bins. Digits 3 and 9 have comparably higher `x4y6` CLIQUE values, while digits 5, 6, and 9 have comparably higher `x3y6` CLIQUE values.

Additionally, the CLIQUE values of `x3y6` appear to be a good metric for distinguishing between the groups of digit 1. The top-right group has strictly high CLIQUE values for `x3y6`, while the larger main group tends to have lower CLIQUE values. This shows that `x3y6` or its CLIQUE values may be a useful metric for parsing out this subgroup structure. See Appendix E.1 and E.2 for additional results.

## 5    Conclusions

CLIQUE offers a unique perspective on local importance, highlighting conditional or locally dependent information that can complement and add new insights to those provided by SHAP, LIME, and ICI.

### 5.1    Limitations

Despite CLIQUE's contributions, some limitations should be acknowledged. We emphasize that the "conditional" effects captured by CLIQUE are interaction-based rather than probabilistic, and should not be interpreted as estimating formal conditional distributions. However, potential connections between CLIQUE and formal conditional expectation frameworks remain an open direction for future theoretical development.

We also acknowledge that quantile-grid replacement creates out-of-distribution inputs in many settings (Hooker et al., 2021). This is not inherently negative and can often provide very useful information in

many applications (Chen et al., 2020; Bladen & Cutler, 2024), but it can limit metrics to analyzing shared predictor associations with the response, rather than individually isolated predictor associations with the response, especially in high correlation settings.

We also note that CLIQUE depends substantially on several modeling and data-related factors. If the underlying predictive model is poorly calibrated or improperly specified, then CLIQUE, SHAP, LIME, and ICI all risk being adversely affected. Additionally, because CLIQUE relies on cross-validation, its performance may be sensitive to poor partitioning strategies, particularly in small or imbalanced datasets. For these reasons, we encourage analysts to prioritize appropriate model specification and cross-validation design both to improve predictive performance and to mitigate potential instability in local importance estimates.

## 5.2 Discussion

While this paper focuses on comparing the values of the local importance measures themselves, we acknowledge that **P6**, computation speed, is often a key consideration. Time-cost comparison plots and analyses can be found in Appendix F. From this analysis we found that SHAP and CLIQUE are noticeably faster than LIME and ICI. Additionally, CLIQUE scales linearly with more observations, which is better than the quadratic scaling found in SHAP. Conversely, SHAP scales roughly independently with respect to the number of features, which is superior to the linear scaling of CLIQUE. Thus the current version of CLIQUE is competitive with SHAP, LIME, and ICI, with room for further improvements.

In assessing the local importances, we observe that under default hyperparameters, LIME values consistently capture marginal information exclusively. This is not surprising, since LIME tends to be focused on explaining individual predictions, and is rarely applied globally as we have done here. ICI also captures marginal information when considering the importance of variable $j$ on observation $i$, as we have done throughout this paper. SHAP provides a decomposition that blends marginal and conditional information, but the conditional structure is often biased by the dominant signal. In the conditional/interacting predictor settings considered here, LIME, ICI, and SHAP often assign nonzero importance in regions where CLIQUE returns values near zero. From the perspective of **P1**, these attributions may be viewed as reflecting broader marginal effects rather than purely local conditional importance.

In contrast, CLIQUE is explicitly conditional, measuring how a variable value affects prediction error rather than the prediction itself. Near-zero values indicate little effect on individual error, while positive values suggest the variable meaningfully reduces it. By focusing on conditional error reduction, CLIQUE can reduce the tendency to assign importance in regions where local prediction error is largely insensitive to a variable. Global assessments can be constructed from the centers and spreads of CLIQUE values. We note that if a variable is more important in a particular region of the feature space, the center and the spread of its CLIQUE values will generally be higher in that region.

In summary, we have provided the structure for a new and novel local importance algorithm. We have shown many applications and contributions of CLIQUE, including its powerful use for studying conditional or locally dependent importances, mitigating false-positive discovery, improvement over permutation-based methods, and direct use in multi-class classification problems. This research offers many future opportunities for expansion into global importance methods and improving permutation-based algorithms generally.

**Code & Data Availability**

The CLIQUE R and python functions, along with code used to generate the data and figures in this paper can be found at: `https://github.com/KelvynBladen/CLIQUE`

**Acknowledgments**

The authors thank Brennan Bean for his review of this manuscript. We are also grateful to the anonymous reviewers and editors of Transactions on Machine Learning Research for their careful evaluation and constructive feedback. Their suggestions helped strengthen both the presentation and quality of this research. The authors also acknowledge OpenAI's ChatGPT (GPT-5) for assistance with language editing to improve the readability of the manuscript. While its help was greatly appreciated, all research contributions, analy-

ses, and conclusions are solely the responsibility of the authors. This research was supported in part by NSF grant 221232. The authors are solely responsible for the content of this work. The views expressed do not necessarily reflect those of the funding agencies. The funders had no involvement in the study design, data collection and analysis, decision to publish, or manuscript preparation. The authors declare no competing interests.

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

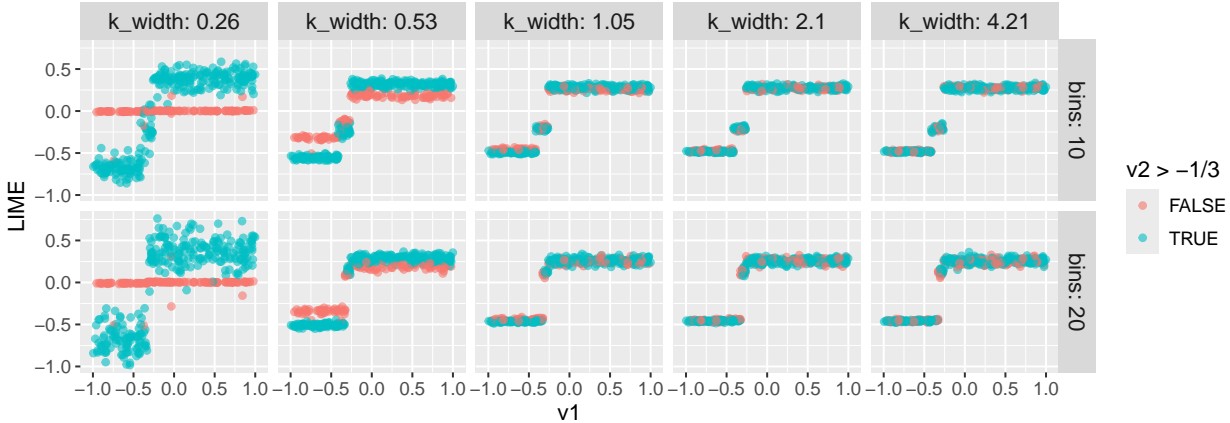

Figure 18: Sensitivity analysis for LIME importances across kernel widths and number of bins in the AND-gate simulation. $v_1$ values are on the x-axis, while LIME values are on the y-axis. Kernel widths were defined by multiplying the median pairwise distance in the data by $\{0.2, 0.4, 0.8, 1.6, 3.2\}$. Smaller kernel widths more clearly recover the imposed conditional structure. Reducing the number of bins also yields modest variance reduction in the resulting explanations.

## A  LIME Hyperparameter Sensitivity Analysis

Figure 18 illustrates the sensitivity of LIME explanations to both the kernel width and number of bins parameters. The kernel width controls the locality of the surrogate model fit, with smaller values placing substantially more weight on observations near a target point. In this simulation setting, smaller kernel widths more clearly recover the imposed conditional structure.

Importantly, however, kernel width requires manual tuning and substantially affects the resulting explanations. At present, there is no widely accepted optimization criterion for selecting these parameters in real-world settings where the underlying conditional structure is unknown. This analysis motivates caution when interpreting default LIME explanations and offers some brief empirical insights regarding how different kernel width values may affect the LIME importances in localized explanation methods.

## B  Regression Loss Comparison

Here we offer a brief comparison of basic loss functions for the Regression Interaction data found in Section 3.3. Figure 19 illustrates the influence of the selected loss function on the resulting CLIQUE values. While both the squared-error and absolute-error formulations recover the underlying interaction structure, the squared-error loss produces substantially larger differences in importance when $v_3 > 0$ due to its increased sensitivity to large residuals. In contrast, the absolute-error loss yields noticeably more stable local importance estimates while still preserving the conditional relationship induced by $v_3$. This highlights that CLIQUE explanations are inherently tied to the chosen loss function, and that different losses may emphasize different aspects of model behavior and predictive error. Throughout the paper, we employ absolute-error loss functions due to their increased robustness.

## C  Extended Results on High Dimensional Data

Here we show extended results from the Higher Dimensional Correlated dataset from Section 3.4. We maintain all aspects of the design except for adjusting the correlation to be $\rho = 0.25$ (see Figure 20) and then $\rho = 0$ (see Figures 21). Thus variables 11-100 have less predictive information for the response compared to the original setting. For $\rho = 0.25$, both LIME and CLIQUE reflect the imposed conditional structure,

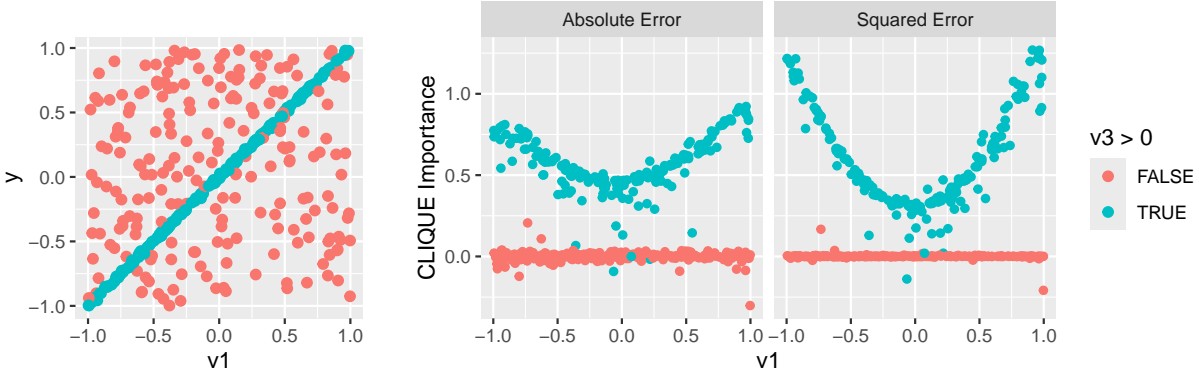

Figure 19: Scatterplots of the response variable (left-panel) and two CLIQUE importances (right-panels) versus $v_1$ for the Regression Interaction data. Each plot is colored according to whether $v_3 > 0$ (see Equation 6). The absolute error yields CLIQUE values which are far more robust to extrema than the squared error.

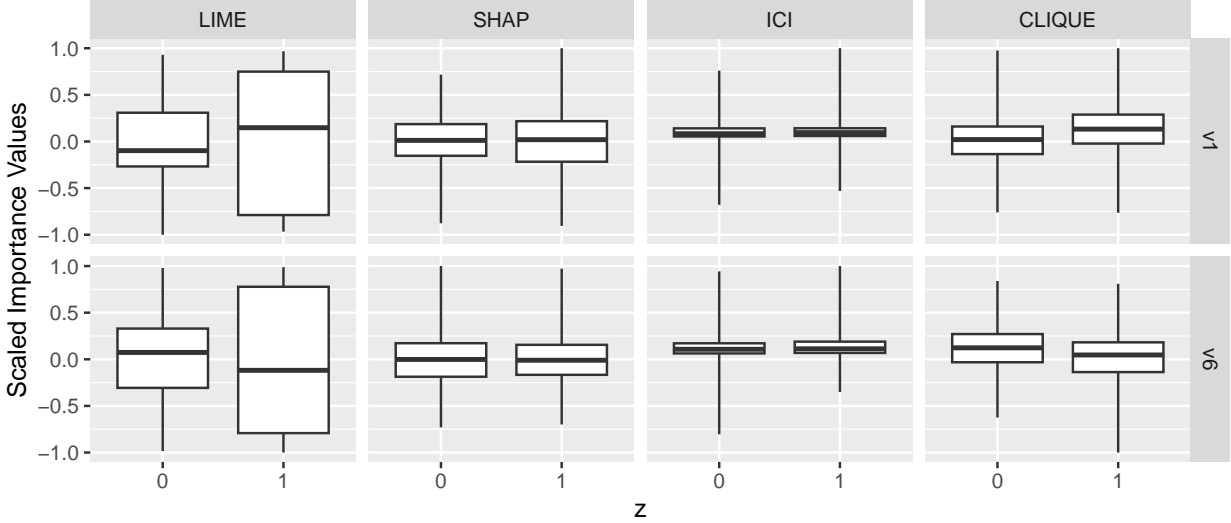

Figure 20: Distributions for $v_1$ and $v_6$ importances across $z$ using the dataset described in Section 3.4 with Equation 7 with $\rho = 0.25$. Importances are scaled by their maximum absolute value within each method. Classically defined outliers are included in whiskers. Both CLIQUE and LIME reflect the conditional structure, although the variance of the LIME values is larger.

although the variance for the LIME values is much higher. For $\rho = 0$, only CLIQUE reflects the correct structure.

In addition to varying the correlation, we also reset the correlation to $\rho = 0.5$ and impose a marginal effect for $z$ (see Figure 22). To impose such an effect, we adjust Equation 7 to be as follows:

$$y = \begin{cases} 3(v_1 + v_2 + v_3 + v_4 + v_5) + 3 + \epsilon, & \text{if } z = 1 \\ 3(v_6 + v_7 + v_8 + v_9 + v_{10}) - 3 + \epsilon, & \text{if } z = 0. \end{cases} \tag{8}$$

Again, only CLIQUE reflects the correct conditional structure.

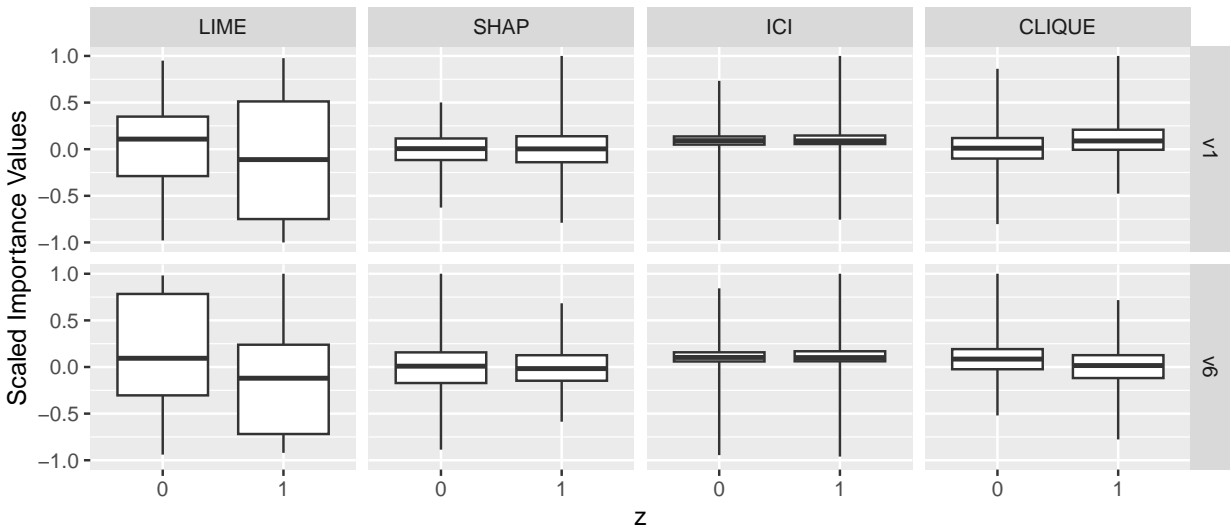

Figure 21: Distributions for $v_1$ and $v_6$ importances across $z$ using the dataset described in Section 3.4 with Equation 7 with $\rho = 0$. Importances are scaled by their maximum absolute value within each method. Classically defined outliers are included in whiskers. CLIQUE most closely reflects the correct conditional structure.

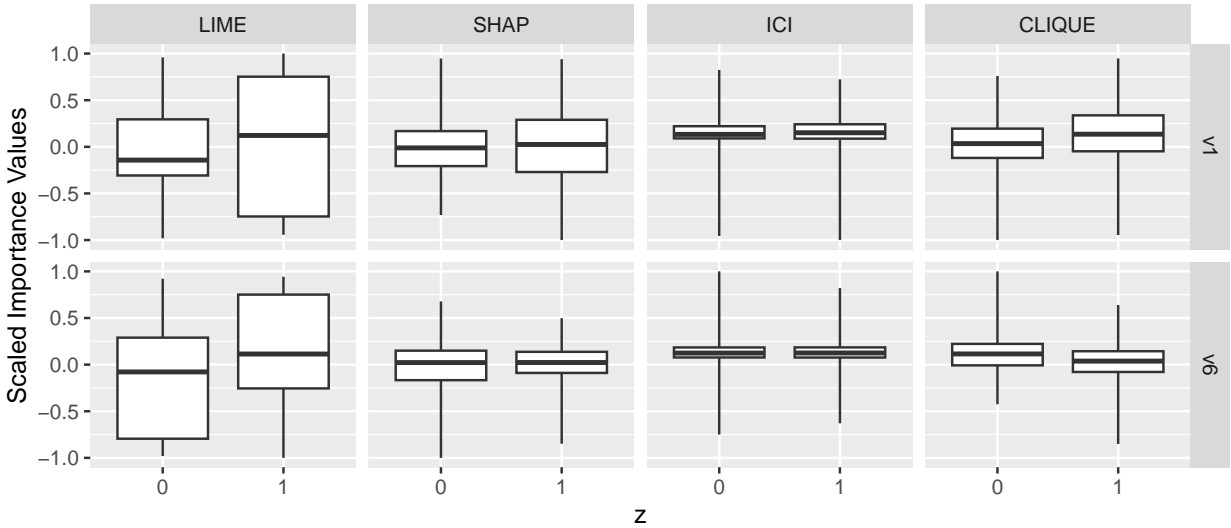

Figure 22: Distributions for $v_1$ and $v_6$ importances across $z$ using the dataset described in Section 3.4 with $\rho = 0.5$ and Equation 8. Importances are scaled by their maximum absolute value within each method. Classically defined outliers are included in whiskers. CLIQUE again most closely reflects the correct conditional structure.

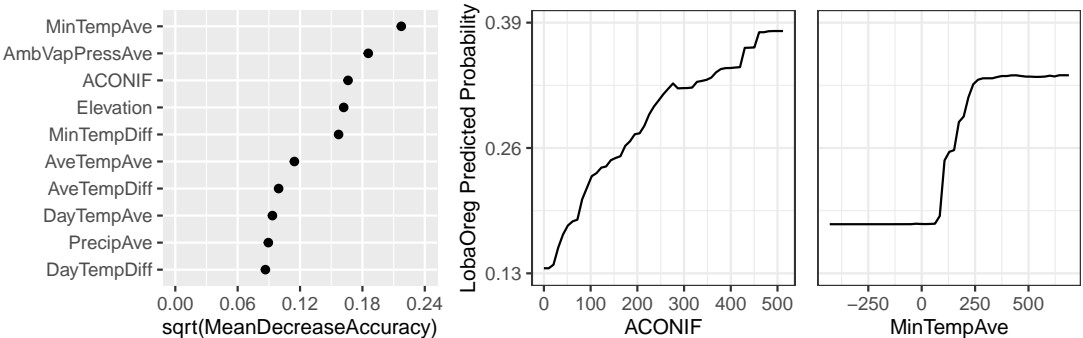

Figure 23: Global importances (left) and PDPs (right) for top features in the Lichen data.

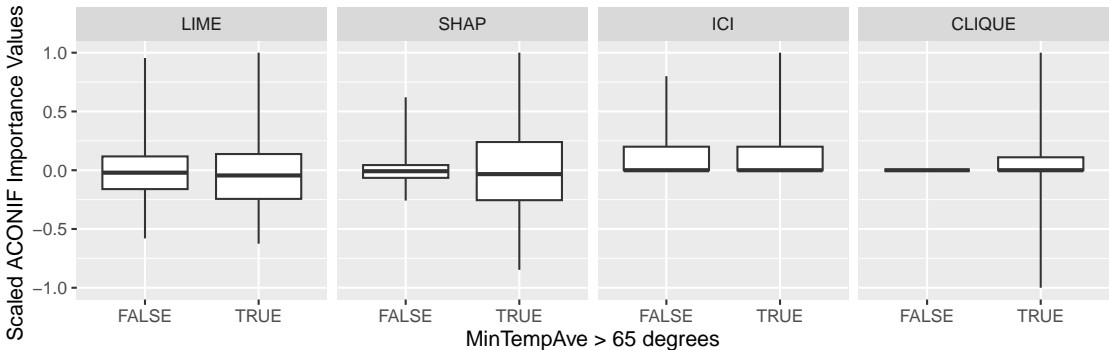

Figure 24: Distributions of Random Forest ACONIF importances for different MinTempAve thresholds. Importances are scaled by their maximum absolute value within each method. Classically defined outliers are included in whiskers. The CLIQUE values show the greatest contrast between the two regions compared to the other methods, followed in order by SHAP, LIME, and then ICI.

## D    LICHEN Data with Random Forest

To offer some comparison across different models, we show results for the Lichen data again, but for a Random Forest model using all available features, and extract local importance values from it. Results for permutation-based global variable importances (Breiman, 2001) and selected PDPs are shown in Figure 23. In comparison to the XGBoost results, MinTempAve (minimum temperature) and ACONIF (age of conifers) are less pronounced, but still among the most important variables globally for classifying the presence of Lichen. Descriptive analyses show that MinTempAve is highly collinear with AmbVapPressAve ($r = 0.997$) and Elevation ($r = -0.978$), while being almost orthogonal to ACONIF ($r = -0.116$). We focus on MinTempAve for direct comparison with the results in Section 4.2, but also acknowledging that very similar patterns are likely to exist for AmbVapPressAve and Elevation in this model. The ACONIF PDP again shows a steady upward trend (this time more linear), while MinTempAve again exhibits a steep sigmoidal trend.

Following the same procedure for the XGBoost model, we aggregate ACONIF importances based on whether MinTempAve is below 65. The resulting boxplots in Figure 24 show that ACONIF has almost exactly zero CLIQUE importance when minimum temperatures are lower. The SHAP, LIME, and ICI value distributions in Figure 24 align nicely with those of Figure 13, with the primary notable difference being an even more distinct difference in ACONIF Importance values across the two temperature groups.

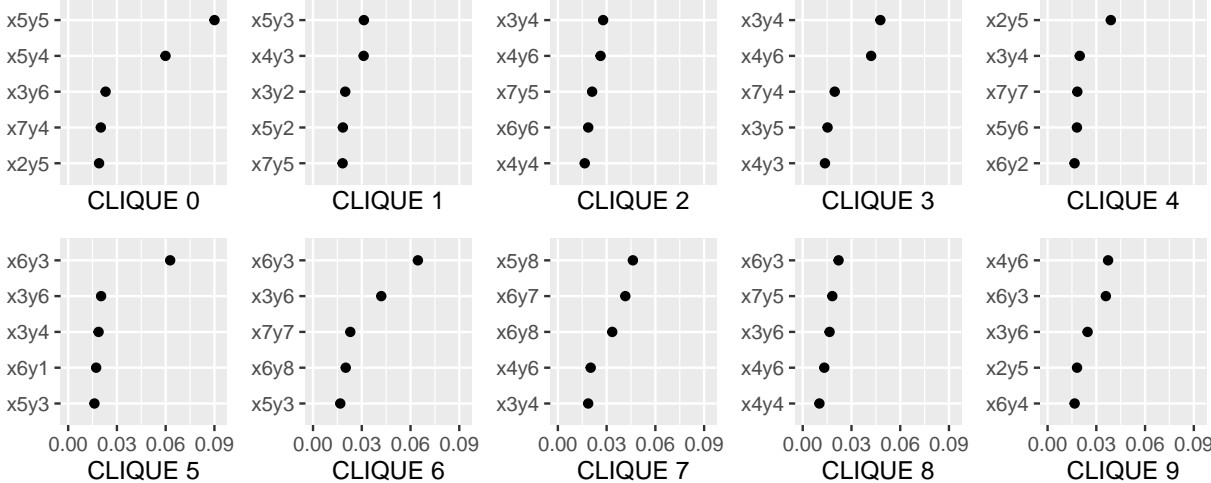

Figure 25: Plot showing top 5 predictor variables based on mean CLIQUE values for each digit class. Using the CLIQUE values, digits can be identified that benefit from globally important pixels of `x4y6` and `x6y3`.

# E    Additional MNIST Results

Here we show extended results from the MNIST digit classification dataset.

## E.1    MNIST Results by Class

In Section 4.3, we explored CLIQUE importances for MNIST pixel groups. We now take a different approach of assessing top pixels across each digit class. This is done by calculating the mean CLIQUE value for each pixel across each class (**P7**). We collect the top five pixels for each class and show them in Figure 25. From these results, we see that no pixels containing `x1` or `x8` are included. This matches trends in Figure 14, where we see those two columns of pixels are always zero for each of the example digits. We also observe that digits 3 and 9 share `x4y6` as a top pixel. In Section 4.3, we showed that the distribution of `x4y6` CLIQUE values was elevated substantially for these two digits in comparison to the others.

Meanwhile, digits 5 and 6 both share `x6y3` as their most important pixel. This was followed by `x3y6`, which we discuss in Section 4.3 as a strong discriminator for these digits. `x6y3` does not seem like a powerful pixel for splitting 5s and 6s, since both digits tend to have low values for this pixel. However, it is very effective for splitting 5s and 6s into their own cluster, away from the remaining digits, which generally have high values for `x6y3`. A noteworthy takeaway is that CLIQUE importance is focused on how a value compares to the whole distribution of that variable. If eight classes have a high pixel value, while two have a low value, that pixel will be very important for the two different classes, but not as important for the other eight.

We can see this in digit 0, which has a very high CLIQUE value for `x5y5` and `x5y4`. Despite this, both pixels never show up in the top five pixels for any other digit. A likely reason is that these pixel values are low for 0, but higher for all other digits on average. Thus, digit 0 is anomalous with respect to these pixels, making them very effective for distinguishing 0s from the rest of the digits.

## E.2    MNIST Samples

In addition to examining top pixels across digit classes, we further explore the two PHATE clusters for digit 1. We select central points from these clusters (denoted A and B, respectively), visualize them, and compare their top CLIQUE pixels in Figure 26. CLIQUE highlights pixels that distinguish these digit 1 samples from other digits. Although the images are visually distinct (A includes an underline whereas B does not), they

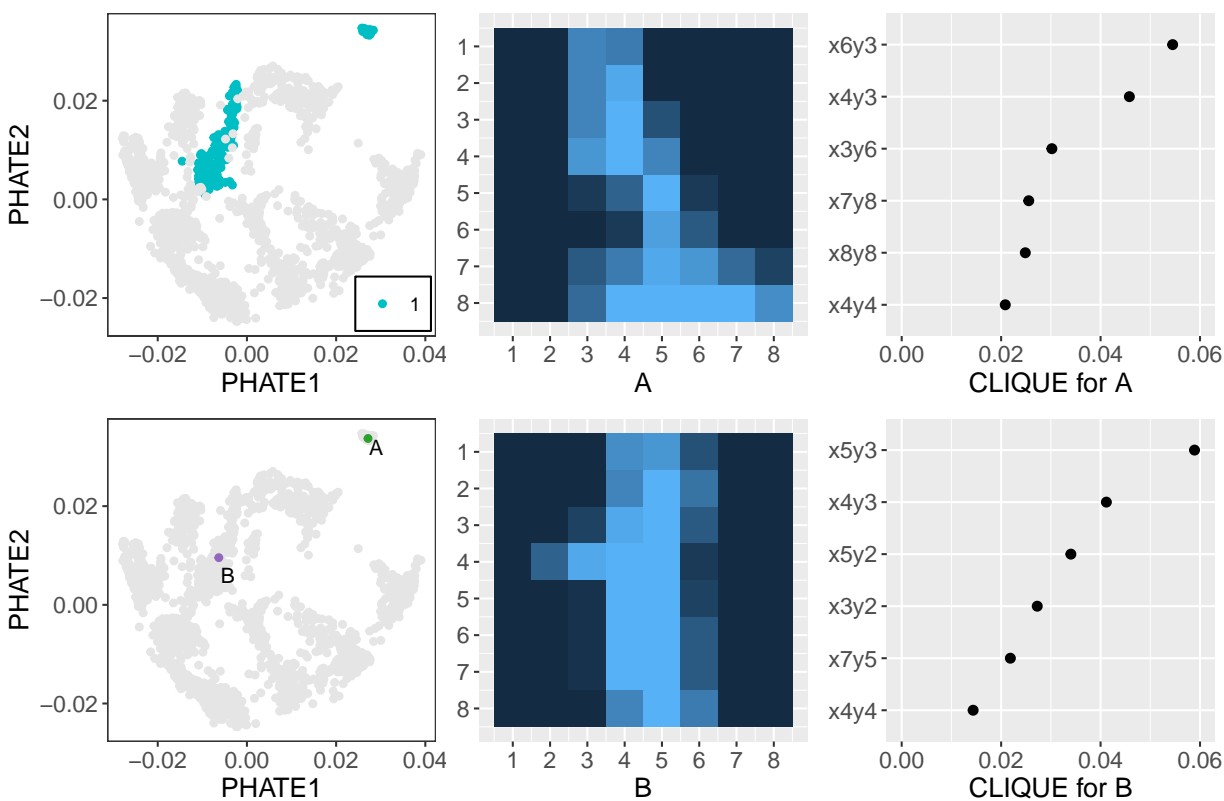

Figure 26: Plot showing samples from the two digit 1 PHATE clusters. Top-Left: PHATE colored by digit 1 class. Bottom-Left: PHATE colored by samples A and B. Middle: Tile plot of A and B. Right: Top 8 pixels from CLIQUE values for A and B.

share some top-ranked pixels, `x4y3` and `x4y4`. Both pixels have high values for the two samples, indicating similar local importance patterns despite morphological differences.

Perhaps more notable are the differences that emerge. Sample B has comparatively high values for `x5y3`, `x7y8`, and `x8y8`, increasing their importance for that observation, whereas sample A does not share these intensities, rendering these pixels less influential for its classification. Conversely, sample A has distinctly low values for `x6y3` and `x3y2`, along with a high value for `x5y2`. These intensities increase the corresponding importances for sample A, whereas the same pixels are not as extreme for sample B, implying that they are less influential for predicting its label.

## F   Time Cost Comparison

Here we perform a computational time comparison between CLIQUE, SHAP, ICI, and LIME. We use the MNIST dataset with images aggregated to a 6x6 pixel resolution (Rene, 2025) to test the effects of increasing both the number of observations and the number of features on computational time. To ensure compatibility with both SHAP and LIME (which are adapted natively only to binary classification), we limited the dataset to just two digit classes: 3 and 8.

For this experiment, we use a Random Forest with default hyperparameter settings. We implement tree-SHAP using its default parameters, LIME with the number of bins increased to 10, ICI with the number of permutations reduced from 50 to 25, and CLIQUE with 25 quantile grid points. Reported computation times include the fitting of five cross-validation models for CLIQUE, parallelized across five cores. Consequently,

model training times are included for each respective method for comparison sake. These experiments were all conducted on a Windows computer with an AMD Ryzen 7 4700U 2.00 GHz processor, featuring 8 cores and 16 GB of RAM.

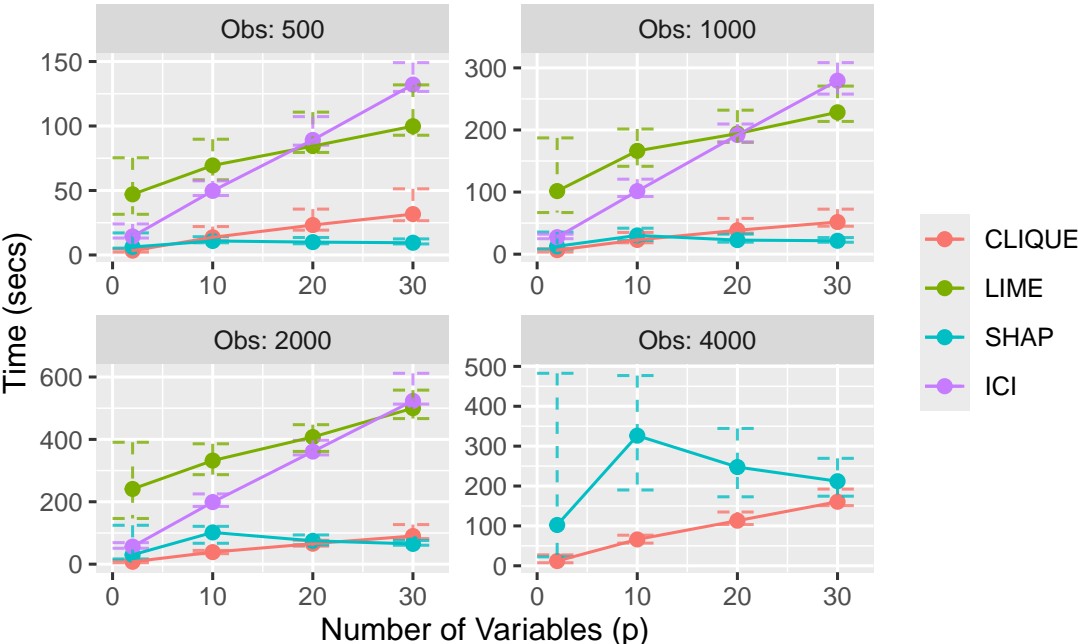

Figure 27: Plot showing computation time results as features increase, faceted by observations. The points and error bars represent the minimum, median, and maximum times from 20 Monte Carlo simulations.

Figure 27 shows the computational time in seconds for all four methods as a function of the number of features and for different sample sizes. From these results, we observe that both SHAP and CLIQUE are significantly faster than LIME and ICI (**P6**). Additionally, the figure reveals that the SHAP runtime is approximately non-increasing with respect to the number of features, although the variance can be high. In contrast, the CLIQUE runtime increases linearly with the number of features but is more stable.

Figure 28 shows the same results with the computational time as a function of sample size across different numbers of features. From these graphics, we observe that the SHAP runtime generally increases quadratically with the number of observations, while the CLIQUE runtime remains linear.

These results suggest that CLIQUE scales better with increasing observations, while SHAP scales better with more features. This aligns with the expectations set by the authors of TreeSHAP (Lundberg et al., 2018), who describe a fast SHAP implementation with time complexity $O(TLD^2)$, where $T$ is a model hyperparameter, $L$ is the maximum number of tree nodes (which is largely dependent on the number of observations $n$), and $D = log(L)$. Notably, the SHAP time complexity does not depend on the number of features $p$, implying that computational cost remains roughly constant across different values of $p$. While $p$ does influence $L$, this relationship is not monotonic and is significantly weaker than the effect of $n$.

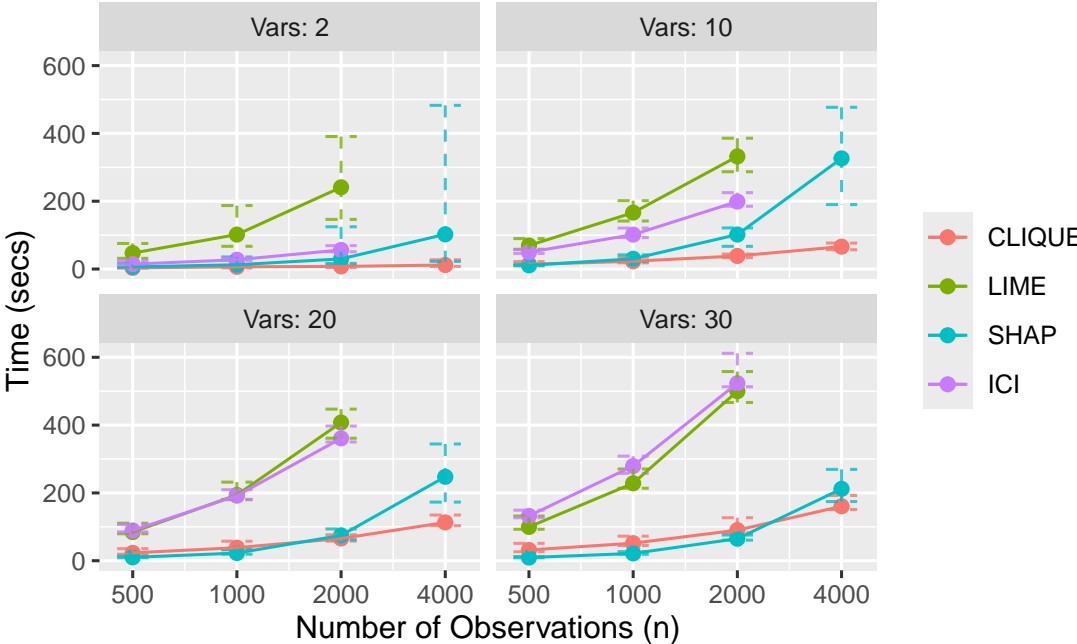

Figure 28: Plot showing computation time results as the number of observations increase, faceted by variables. The points and error bars represent the minimum, median, and maximum times from 20 Monte Carlo simulations.

