# OpenReview forum: "Conditional Local Importance by Quantile Expectations"
_TMLR — Accepted by TMLR_

### Review · Reviewer_jHyc · 2026-04-15

**Summary Of Contributions:**

This paper introduces CLIQUE (Conditional Local Importance by QUantile Expectations), a model-agnostic method for local variable importance in machine learning interpretation. The core idea is to measure how perturbing a single feature, by substituting its value with quantile-grid replacements, changes cross-validated prediction error for individual observations. Key contributions include:
(1) a formal framework with eight desirable properties (P1–P8),
(2) a mathematical proof that CLIQUE assigns exactly zero importance to locally invariant features,
(3) empirical comparisons against LIME, SHAP, and ICI on simulated and real datasets,
(4) native support for multi-class classification without one-vs-all decomposition
(5) favorable computational scaling with respect to sample size relative to SHAP.

Strengths include a clear conceptual motivation and the feature-invariance guarantee (Proposition 1). Weaknesses include limited theoretical analysis beyond Proposition 1, reliance on cross-validation introducing computational and methodological complexity not fully analyzed, and comparisons that may not represent SHAP and LIME at their best configurations.

**Audience:**

Yes

**Audience Explanation:**

Yes. Interpretable machine learning and explainability are active, high-interest areas in the TMLR community. The specific problem addressed is a genuine and underappreciated limitation of SHAP and LIME. The multi-class classification extension and the quantile-replacement stability argument both address real practical needs.

**Broader Impact Concerns:**

No impact concerns.

**Claims And Evidence:**

Yes

**Claims Explanation:**

The claims are supported by evidence but there are some critics to address.
The primary empirical claim, that CLIQUE better suppresses false-positive importance in locally invariant regions, is supported consistently across three simulated experiments (AND gate, Corners, Regression Interaction) and quantified via FP-MAE in Table 1, where CLIQUE outperforms competitors by a substantial margin. The real-data applications (Concrete, Lichen, MNIST) are illustrative and well-chosen.
Cross-validation design is underspecified. CLIQUE requires fitting multiple CV models, yet details about fold counts, stratification, and sensitivity to CV choices are not provided. This matters for both reproducibility and the validity of P8 (out-of-sample evaluation).
The paper compares against specific R package implementations (treeshap, lime) with default settings. For instance, TreeSHAP is designed for tree-based models and has known strengths precisely in the interaction-detection settings tested here. Whether more carefully tuned or kernel-SHAP configurations would fare better is not explored.

**Requested Changes:**

1. Fully specify the cross-validation procedure. State the number of folds, how folds are constructed (especially for classification), and provide a sensitivity analysis showing how CLIQUE values change under different CV configurations. This is essential for reproducibility and for substantiating P8.

2. Broaden the SHAP/LIME comparison. Include kernel SHAP (model-agnostic, not tree-specific) and at minimum one sensitivity analysis varying LIME's kernel width or perturbation strategy. The current comparison risks conflating SHAP's implementation limitations with its conceptual limitations.

3. Clarify the relationship to ICI more precisely and provide a section on the state of the art. The paper states CLIQUE "shares algorithmic similarities with ICI" and describes it briefly. A section for the state of the art for the model you cite (LIME, SHAP ad ICI) could help to understand better the claim of the paper i.e. the limitations of these models and the advantages of the one you proposed.

4. Discuss sensitivity to the quantile grid size M more thoroughly. Figure 2 suggests values stabilize around M=15–20, yet M=25 is used throughout. A brief ablation across datasets would strengthen the practical guidance.

---

> ### Author Response · Authors · 2026-04-29
> **Initial Response to jHyc with Revision Plans to Accommodate**
>
> We thank the reviewer for offering their compliments of our work and ideas for enhancing
> it further. We are providing a quick response for now in case the reviewer wants to provide
> any feedback on our revision plans. We will provide more details about the actual revisions
> and the results of the additional experiments along with our revised manuscript within the
> next week and a half.
>
> Requested Changes
>
> 1. We agree and thank the reviewer for catching this oversight. We will provide the CV
> details in our revision and we have begun a simple CV sensitivity analysis for one of
> our simulated datasets to help increase our study of this issue.
> 2. We thank the reviewer for these helpful suggestions. We agree that a sensitivity analysis
> varying one of LIME’s hyperparameters is warranted and are working on that now. We
> also believe our revisions to the comparative language between CLIQUE, SHAP, and
> LIME will improve these issues.
> Regarding other SHAP versions, in our ANN experiment on the 2 Corners dataset,
> SHAP values were computed using shap.Explainer in python, which in this setting
> leverages a DeepExplainer-style approach rather than a tree-specific implementation.
> We agree that this distinction was not sufficiently provided in the manuscript and
> will clarify the specific SHAP variant used in the revision. For our tree-based models
> (which are the vast majority), we used TreeSHAP due to its computational efficiency,
> exactness for these model classes, and its strengths in interaction detection settings.
> Since KernelSHAP is an approximation and computationally slower, we do not expect it
> to have a better performance compared to TreeSHAP. Given these facts and the variety
> of other experiments we have agreed to perform for this revision, we would prefer to not
> prioritize an experiment using kernelSHAP. Regardless, in the revision, we will provide
> a clearer explanation of all of these points. But if the reviewer strongly believes that we
> should include an experiment with kernelSHAP in addition to the other experiments
> we have agreed to do, please let us know as soon as possible.
> 3. We thank the reviewer for this point and agree that an extended discussion of the
> current state of the art methods would improve the paper. In particular, we will better
> clarify the relationship between CLIQUE and ICI. We will also dedicate more content
> in the introduction to provide greater background for LIME, SHAP, and ICI.
> 4. We agree that further discussion of CLIQUE’s sensitivity to the number of quantiles
> is warranted. We note that while values stabilize around M=15-20 in Figure 2, we
> selected 25 to provide us with some "wiggle room" to account for potential variability.
> We will include a discussion of this in the revision. We also will include an ablation of
> the number of quantiles for the regression interaction dataset to further strength this
> guidance.

---

> ### Author Response · Authors · 2026-05-12
> **Complete Response to jHyc**
>
> We again thank the reviewer for offering their ideas for enhancing our work. We have fully carried out the revisions we discussed previously and we believe the paper has been greatly improved. We note that we have used a “track changes" approach to highlight the specific changes to the paper for convenience. We hope that the reviewer finds the revised version of the paper satisfactory.
>
> Requested Changes
> 1. We have provided the CV details in our revision and we have included a simple CV sensitivity analysis for one of our simulated datasets in Section 3.6 to help increase our study of how CLIQUE and local permute are affected by different CV configurations. Overall, CLIQUE is stable to the number of CV partitions, especially when 5 or more partitions are used.
> 2. We have included a sensitivity analysis varying some of LIME’s hyperparameters. See Appendix A. We found that smaller kernel widths more clearly recover imposed conditional structures. However, the kernel width requires manual tuning and there is
> no widely accepted criterion for doing so when the underlying conditional structure is unknown. Thus we chose to keep the default settings for LIME in the main paper as that is the most realistic comparison. But we have updated our comparative language
> between CLIQUE, SHAP, and LIME to reflect these new findings. We have also revised our text to clarify our specific uses of deepSHAP and treeSHAP.
> 3. We have expanded our discussion of the current state of the art local importance methods throughout the Introduction. We have better clarified the relationship between CLIQUE and ICI and provided a greater background for LIME, SHAP, and ICI.
> 4. We have included an ablation of the number of quantiles for the regression interaction dataset to better demonstrate robustness to quantiles. See Section 3.6. We have also updated the text based on this to better explain our selection of 25 quantiles.

---

### Review · Reviewer_9m9p · 2026-04-16

**Summary Of Contributions:**

The paper proposes CLIQUE (Conditional Local Importance by Quantile Expectations), a new model-agnostic local feature importance method. The core idea is to measure how much the prediction error changes when that feature is replaced across a quantile grid, while holding the other features fixed. This is presented as an alternative to local explanation methods such as LIME, SHAP and ICI, with the explcit goal of better capturing locally dependent or interaction-driven importance.

Main contributions are:
1) it introduces the method formally and states a set of desired properties, including feature invariance, stability, model-agnosticism, applicability to multiclass classification, and aggregation to subgroup/global summaries. It also includes a simple proposition showing that if the model output is invariant to a feature for a given observation, then the corresponding CLIQUE importance is zero.
2) it provides empirical evidence on synthetic datasets with known local interaction structure. In the AND-gate, Corners, and regression-interaction settings, CLIQUE appears to recover near-zero importance in regions where a variabe should be irrelevant, while LIME, SHAP and ICI often assign nonzero importance.
3) the paper presents real-data illustrations on concrete strength, lichen classification and MNIST. These examples are used to argue that CLIQUE yields more conditionally informative subgroup patterns and that it extends naturally to multiclass problems because it is defined on loss rather than class-specific additive decompositions.

- Strengths: The paper addresses a real gap in local explanation methods by distinguishing true local dependent relevance from marginal or globally dominant effects. The method is model-agnostic and the synthetic experiments are well chosen because ground-truth local relevance is known. The multiclass setup is also potentially useful.

- Weaknesses: The evidence is promising but still somewhat limited. Much of the case relies on qualitative plots and a few hand-crafted simulations. Experiments on real data need to be more validated.

**Additional Comments:**

I found the paper interesting and promising, with the synthetic experiments being its strongest part, but it request some changes to be mature for TMLR.

**Audience:**

Yes

**Audience Explanation:**

Yes. The paper addresses a problem of broad interest to TMLR's audience, how to obtain local feature importance measures that capture interactions and context-dependent relevance rather than merely marginal importance, which is directly relevant to researchers working on interpretable ML and explanation methods.

**Broader Impact Concerns:**

I do not see major ethical concerns specific to the proposed method itself.

**Claims And Evidence:**

Yes

**Claims Explanation:**

- The paper provides clear and reasonably evidence for its central empirical claim that CLIQUE can better suppress false-positive local importance in settings with strong local interaction structure. This is supported by three synthetic experiments where the true local relevance pattern is known, and by the FP-MAE summary table where CLIQUE substantially outperforms LIME, SHAP, and ICI. The feature-invariance proposition also supports one important part of the method’s design.

- However, some claims are not well supported. The strongest claims about recovering conditional/local structure and reducing spurious importance are supported mainly on synthetic data in relatively stylized low-dimensional setups, with respect to more realistic correlated-feature settings. The real-data sections are interesting and show plausible subgroup differences, but they remain mostly illustrative rather than validating

**Requested Changes:**

- The concrete, lichen, and MNIST examples are interesting, but they would be stronger with more systematic quantitative evaluation rather than mostly qualitative interpretation of plots and subgroup summaries.
- The paper should include broader and more rigorous comparisons, especially higher-dimensional settings
- The paper explicitly says that “conditional” is not used in a formal probabilistic sense. This is important and should be made much more prominent, including in the introduction and discussion, because many readers will otherwise infer conditional feature importance in a stronger statistical sense.
- The paper studies sensitivity to the number of quantiles, which is useful, but it would help to report robustness to other design choices: loss function, CV scheme, model hyperparameters and possibly training randomness.
- Statements such as LIME “consistently captures marginal information exclusively” or SHAP failing to capture local interactions should be softened or better qualified.

---

> ### Author Response · Authors · 2026-04-29
> **Initial Response to 9m9p with Revision Plans to Accommodate**
>
> We thank the reviewer for taking the time to review of our paper and provide their
> meaningful suggestions. We greatly appreciate their praise for our work’s strengths and
> believe that by implementing their recommendations, the paper will be greatly improved
> and an important contribution to TMLR. We are providing a quick response for now in case
> the reviewer wants to provide any feedback on our revision plans. We will provide more
> details about the actual revisions and the results of the additional experiments along with
> our revised manuscript within the next week and a half.
>
> Requested Changes
>
> • (Real world datasets) We agree with the reviewer that quantitative evaluations of
> the real datasets would be helpful. However, quantitative comparisons in real-world
> data are difficult since ground-truth is entirely unknown. To further aid our claims
> we are in the process of adding a more complex, simulated experiment (see below).
> However, if the reviewer has any specific suggestions of quantitative evaluations that
> can be applied to the real datasets, we would be very willing to obtain and include
> them.
>
> • (More Comparisons) We agree with the reviewer that the paper could benefit with
> more rigorous comparisons. We are in the process of implementing a regression simulation with known localized effects for 100 features with some multivariate correlation
> imposed, ρ = 0.5. When a binary feature is 1, a linear combination of the first 10
> features will affect the response. When it is 0, a linear combination of the next set of
> 10 features will affect the response. We will include these results in the revision.
>
> • (Use of the word "conditional") We thank the reviewer for their feedback on this
> and will clarify this more in the introduction and discussion to help mitigate the possibility of readers missing this important clarification.
>
> • (Design choice robustness) We also agree that further experiments on this point
> would be beneficial. We are currently running a sensitivity analysis for CV folds to
> include in the revision. See our responses to reviewer gLmc regarding model calibration/hyperparameter tuning, demonstrating different error functions, and monte-carlo
> simulations for training randomness.
>
> • (Soften the language around LIME and SHAP) We agree that this can be improved and will revise the language throughout accordingly. See also our response to
> reviewer gLmc on this point.

---

> ### Author Response · Authors · 2026-05-12
> **Complete Response to 9m9p**
>
> We again thank the reviewer for their suggestions for our work. We have fully carried out the revisions we discussed previously and we believe the paper has been greatly improved. We note that we have used a “track changes" approach to highlight the specific changes to the paper for convenience. We hope that the reviewer finds the revised version of the paper satisfactory.
>
> Requested Changes
>
> • (Real world datasets) Due to the difficulty in obtaining ground truth information for real datasets, we have opted to include a more complex, simulated experiment (see below). We have also provide better quantitative evaluation of simulation performance (see response to gLmc).
>
> • (More Comparisons) We have implemented a regression simulation with known localized effects for 100 features with some multivariate correlation imposed, ρ = 0.5. When a binary feature z is 1, a linear combination of the first 5 features affects the response. When z is 0, a linear combination of the next set of 5 features affects the response. These results are in Section 3.4 of the revision and modeled with an Artificial Neural Network. We note that CLIQUE best reflects the imposed conditional structure
> in comparison with LIME, SHAP, and ICI.
>
> • (Use of the word "conditional") We have clarified our use of the word conditional several times in the text and mentioned linking it to formal conditional derivations as an area of interesting future work.
>
> • (Design choice robustness) We have included a sensitivity analysis for CV folds in Section 3.6 of the revision and have added text to the limitations section regarding model calibration/hyperparameter tuning in an effort to demonstrate robustness to other design choices. Overall, CLIQUE is stable to the number of CV partitions, especially when 5 or more partitions are used.
>
> • (Soften the language around LIME and SHAP) We have revised the text through-out the entire paper to better qualify and substantially soften the competitive language toward LIME and SHAP.

---

### Review · Reviewer_gLmc · 2026-04-24

**Summary Of Contributions:**

The paper introduces CLIQUE, a model-agnostic local variable importance method. The idea is to take each observation, swap out one feature at a time with values from a quantile grid spanning that feature's range, and measure how much the cross-validated prediction error changes. The averaged error difference gives you the importance of that feature for that observation. The main selling point is what the authors call "feature invariance": if a model's prediction for some observation doesn't actually depend on a given feature, CLIQUE will assign it exactly zero importance. They prove this formally and show through simulations that SHAP, LIME, and ICI all fail here by assigning nonzero importance to features that genuinely don't matter locally, which amounts to false-positive attribution. The quantile-grid approach is also shown to have lower variance than permutation-based alternatives, and because importance is defined through error rather than prediction decomposition, the method works natively for multi-class classification without any one-vs-all gymnastics.The paper is well-written and the simulated experiments are effective since you know the ground truth. The visualizations do a good job making the point. On the weaker side, the comparison with SHAP feels somewhat unfair because the two methods are really answering different questions, the real-data experiments are more descriptive than evaluative, and the computational cost of cross-validation isn't fully accounted for in the timing benchmarks.

**Audience:**

Yes

**Audience Explanation:**

Local interpretability is a topic a lot of people care about, both in research and in practice. The observation that SHAP and LIME can assign spurious importance to locally irrelevant features is useful to know even if you ultimately prefer those methods. The quantile-grid idea is simple enough that people could actually implement it, and the multi-class extension via error-based importance fills a real practical gap. I think practitioners working on model diagnostics and researchers working on interpretability would both find something useful here.

**Broader Impact Concerns:**

Nothing major.

**Claims And Evidence:**

Yes

**Claims Explanation:**

The core claim, that CLIQUE gives near-zero importance to locally invariant features while competitors don't, is convincingly demonstrated in the simulated experiments where ground truth is available. The proof of P1 is correct and straightforward, and the FP-MAE numbers in Table 1 put concrete numbers on what the figures show qualitatively.

That said, I have some reservations.

First, the paper frames nonzero SHAP values in the invariant region as "false positives," but SHAP is decomposing predictions, not errors. A feature can meaningfully contribute to a prediction (nonzero SHAP) while having zero error sensitivity (zero CLIQUE). These are different questions, and calling one answer wrong because it disagrees with the other isn't quite right. The paper would be stronger if it positioned CLIQUE as complementary rather than strictly superior.

Second, the timing comparisons in the appendix aren't clear about whether CV model training time is included. CLIQUE requires fitting CV models while SHAP operates on a single fitted model, so if that cost isn't included the comparison is incomplete.

Third, the real-data experiments are suggestive but hard to verify. The concrete and lichen analyses produce reasonable-sounding interpretations (cement matters more at younger ages, conifer age doesn't matter when it's too cold for lichen), but there's no independent way to confirm these are "correct" conditional effects rather than artifacts. The MNIST analysis is stronger since you can visually check pixel-digit relationships.

Fourth, the simulated experiments appear to use single runs without repeated seeds or confidence intervals on the FP-MAE values, which makes it hard to assess reliability. And model diversity is limited; it's mostly Random Forests with one neural network example.

**Requested Changes:**

The most important change is around framing. The paper currently implies SHAP and LIME are simply wrong when they assign nonzero importance to locally invariant features, but these methods answer a fundamentally different question: how does a feature contribute to the prediction versus how does it affect the error. A feature can contribute to a correct prediction while having zero error sensitivity. The abstract, introduction, and throughout the paper should be revised to acknowledge this distinction explicitly and position CLIQUE as complementary to prediction-based methods rather than a replacement. This matters because the strong "they fail, we succeed" narrative is the backbone of the paper, and it's built on a conflation that reviewers and readers will notice.

Second, the computational comparisons need to clearly state whether CV model training is included in the timing results. If CLIQUE requires additional CV models on top of the base model, that overhead should be reported transparently. This is critical because computational competitiveness is listed as a key property (P6).

Third, the simulated experiments should be repeated across multiple random seeds with confidence intervals or standard errors reported for Table 1. Right now it's a single run per scenario, which isn't enough to establish that the FP-MAE differences are reliable rather than lucky.

Fourth, the paper needs a more honest limitations discussion. The quantile-grid replacement creates out-of-distribution inputs when features are correlated. This is the exact same extrapolation problem that Hooker et al. (2021), which the authors cite, raise against permutation importance. How does CLIQUE handle this? Also, importance depends on the CV partition, which could introduce instability, and the paper doesn't address what happens when the underlying model is poorly calibrated.

Beyond these critical points, a few things would strengthen the paper without being strictly necessary. (1) Adding at least one gradient boosting model would make the model-agnostic claims more convincing. (2) A brief discussion connecting CLIQUE to formal conditional independence testing or conditional permutation importance frameworks would help situate the contribution. (3) Some discussion of how the choice of loss function affects interpretation would be useful since the paper uses absolute error throughout but claims generality. (4) The P1-P8 enumeration is verbose. properties like model-agnosticism and computational competitiveness are standard desiderata and don't need multi-paragraph justification, while the genuinely novel ones (P1, the quantile mechanism, multi-class support) deserve more focus.

---

> ### Author Response · Authors · 2026-04-29
> **Initial Response to gLmc with Revision Plans to Accommodate**
>
> We thank the reviewer for their review of our paper and their concrete suggestions. Indeed,
> we believe that after carrying out the suggested revisions, the paper will be greatly improved.
> We are providing a quick response for now in case the reviewer wants to provide any feedback
> on our revision plans. We will provide more details about the actual revisions and the results
> of our additional experiments along with our revised manuscript within the next week and a
> half.
>
> Reservations
> 1. We agree with the reviewer here that the framing of SHAP values isn’t quite accurate.
> See our response below to Requested Change 1.
> 2. Thank you for catching this oversight. See our response below to Requested Change 2.
> 3. We agree. Indeed, this is a great difficulty in real data, since the ground truth on
> variable importances and interactions is unknown. To boost our claims, we have agreed
> to perform a more complex simulated experiment with higher dimensions (see response
> to 9m9p). We believe this and our other revised experiments, combined with our
> existing experiments provide sufficient evidence of our claims. Nevertheless, if the
> reviewer has any suggestions for real datasets with known variable importances and
> interactions, we are willing to run another real-world experiment.
> 4. We agree that multiple monte-carlo runs will enhance our experiments. We are currently
> running these to include in the revision. Regarding model diversity, we primarily chose
> Random Forests due to their general robustness to hyperparameter selection to reduce
> the potential confounding from hyperparameter choices. However, we will rerun one of
> the experiments using an XGBoost model and include the results in the revision.
>
> Requested Changes
> 1. We agree with the reviewer that the framing of SHAP and LIME is not quite right.
> Indeed, we agree SHAP and LIME are approaching the problem from a different per-
> spective (i.e. prediction) and that CLIQUE is complementary. In our revision, we will
> better highlight this distinction and revise the language throughout accordingly.
> 2. We agree with the reviewer and thank them for bringing this oversight to our attention.
> The computational comparisons included the CV cost times for CLIQUE. We will clarify
> this in the revision and provide full details for transparency.
> 3. Indeed, we will include the results using multiple runs to obtain intervals for the FP-
> MAE values.
> 4. We agree that quantile-grid replacement does create out-of-distribution inputs in many
> settings, as Hooker et al.’s paper has pointed out. However, there is currently debate
> among the variable importance research community on the implications of this. For
> example, Chen et al. (2020) and Bladen & Cutler (2024) show that this property of
> permutations is not inherently negative, and can often provide very useful information.
> Thus we view these issues as an open problem that is beyond the scope of this work.
> Nevertheless, as it is a potential limitation, we will expand our discussion of this in the
> revised paper.
> We also agree that the CLIQUE values depend on the CV partition and plan to add a
> simple sensitivity analysis for one of our simulated datasets to help increase our study
> of this issue. We will also discuss this in the limitations.
> As for poor model calibration, we note that like most variable importance methods
> (including LIME and SHAP), CLIQUE provides estimates of the importance of the
> variables for the trained model. Thus if the model is poorly calibrated, the CLIQUE
> values will be influenced by that. We will add a discussion of this in our limitation
> section and especially advise analysts to prioritize proper model calibration prior to
> fitting any local explanability method: CLIQUE, SHAP, or others.
>
> Minor Points
> 1. We agree. See our response to reviewer jHyc about XGBoost.
> 2. We agree and will attempt to provide a brief discussion connecting these idea and
> suggesting this as a meaningful area of future work.
> 3. We have also performed experiments for the Regression Interaction Data using the
> squared error, which strongly favors extreme values (particularly low or high values
> in a variable’s distribution) when assessing importance with CLIQUE. In contrast,
> absolute error is more robust for extreme values. We initially left out these squared
> error results to prevent the paper from becoming too long, but we’re happy to include
> them again, at least in the appendix. We will show these results and include a brief
> discussion.
> 4. Thank you for this suggestion. We will trim some of the standard desiderata paragraphs
> for a more concise narrative, and add more details in the more novel areas as needed.
>
> References
>
> Bladen, K. & Cutler, D. R.: 2024, Assessing agreement between permutation and dropout
> variable importance methods for regression and random forest models, Electronic Research
> Archive 32(7), 4495–4514.
>
> Chen, H., Janizek, J. D., Lundberg, S. & Lee, S.-I.: 2020, True to the model or true to the
> data?, arXiv preprint arXiv:2006.16234.

---

> ### Author Response · Authors · 2026-05-12
> **Complete Response to gLmc**
>
> We again thank the reviewer for their valuable input. We have fully carried out the revisions we discussed previously and we believe the paper has been greatly improved. We note that we have used a “track changes" approach to highlight the specific changes to the paper for convenience. We hope that the reviewer finds the following changes satisfactory.
>
> Reservations
> 1. See our revisions for Requested Change 1.
> 2. See our revisions for Requested Change 2.
> 3. We have added a more complex simulated experiment with higher dimensions and correlation structures, to help expand the scope of our experiments to better match real-data structures (see response to 9m9p).
> 4. See our revisions for Requested Change 3 and Minor Point 1. We have used a Neural Network for our new high dimensional experiment to expand the model-agnostic demos. We have also rerun the Lichen dataset using an XGBoost model and included the results in the revision, with the Random Forest results being moved to the appendix.
>
> Requested Changes
> 1. We have revised the text throughout the entire paper to better highlight the distinction between SHAP/LIME and CLIQUE, and substantially softened the competitive language between them.
> 2. We have clarified in the text that the computational comparisons included the CV cost times for CLIQUE and provided greater details for transparency.
> 3. We have updated Table 1 to include results from 50 MC runs to obtain means and intervals for the MAE values with respect to 0.
> 4. We have added a limitation section and provided a far more comprehensive discussion of relevant limitations. We have specifically included the request to address out-of-distribution inputs in our limitations. We have mentioned CV partitions and added
> a simple sensitivity analysis to the number of CV folds for the AND GATE data in Section 3.6: Effect of Parameters on Permutation vs. Quantile Replacement. We have also mentioned model calibration in the limitations section, advising analysts to
> prioritize proper model calibration prior to fitting any local explanability method.
>
> Minor Points
> 1. We have employed a XGBoost model for the Lichen data. Furthermore, we have moved the Random Forest Lichen results to the Appendix for comparison between models. See our response to reviewer jHyc about XGBoost.
> 2. We have provided a brief discussion regarding formal conditional importance and suggesting this as a meaningful area of future work.
> 3. We have performed a comparison of squared error and absolute error loss functions for the Regression Interaction Data and added these results to the appendix along with a brief discussion explaining why we default to absolute error. We also mention this
> briefly in the main text.
> 4. We have trimmed some of the standard desiderata paragraphs for a more concise narrative and added more details in the more novel areas as needed.

---

### Author Response · Authors · 2026-05-09
**Progress Update for Reviewers**

Given the timeline associated with this process, we wanted to provide a brief progress update regarding the reviews and request a few additional days for the revision.

At this stage, all requested experiments have been completed and their results have been generated. Additionally, the vast majority of requested revisions to the manuscript text have already been incorporated. The primary remaining tasks are completing the write-up for two of the six additional experiments. We expect to complete these remaining components early next week and provide a revised version of the manuscript incorporating your feedback at that time (about May 12). We are aware that formal reviewer decisions are not due until around May 18-22, but if decisions can be submitted now, we would greatly appreciate a short extension in order to finalize the manuscript carefully and thoroughly and appropriately address all requested revisions and additional experiments.

Thank you again for your patience, thoughtful reviews, and consideration.

---

### Decision · Action_Editor_s7Uj · 2026-06-08

**Recommendation:** Accept with minor revision

**Additional Comments:**

I recommend acceptance with minor revisions. The previous revision has substantially addressed the main reviewer concerns.

I would like to ask the authors to carefully clean the manuscript to remove all track-changes or revision artifacts. The final submission should read as a clean manuscript without residual editing marks or duplicated wording. Please also make sure that the submission follows the TMLR tex style of accepted paper.

Moreover, please keep the comparison with SHAP, LIME, and ICI consistently qualified. CLIQUE should be presented as a complementary approach, not as a general replacement. Please soften or qualify absolute statements about the comparisons between CLIQUE and the existing methods.

**Audience:**

Yes

**Audience Explanation:**

The submission addresses local interpretability, feature importance, interaction-dependent explanations, and multi-class explanation settings, all of which are relevant to a meaningful segment of the TMLR audience. The method gives practitioners and researchers a complementary loss-space view of local importance, distinct from prediction-decomposition methods such as SHAP and LIME. All three reviewers marked Audience as Yes, and the final reviews indicate that the paper is sound and relevant to the interpretable ML community.

**Claims And Evidence:**

Yes

**Claims Explanation:**

The revised submission presents CLIQUE, a model-agnostic local variable-importance method that measures observation-level changes in prediction error under quantile-grid feature replacements. The central claim is that CLIQUE is well suited to identifying locally invariant or interaction-dependent feature relevance in the model-loss space, which is supported by the extensive experiments in the paper. Moreover, the authors strengthened the revision with many additional experimental results and sensitivity analyses, which collectively respond well to the reviewers' earlier concerns.

The review process supports this assessment: all three reviewers marked Claims And Evidence as Yes after revision, with final recommendations of Accept, Accept, and Leaning Accept. I find the submission's main claims supported by sufficiently accurate, clear, and convincing evidence for TMLR acceptance.

---

> ### Author Response · Authors · 2026-06-11
> **Response to Action Editor**
>
> We thank the Action Editor for their decision to accept our manuscript! We have worked to accommodate all of your requests.
> - We have removed all track changes and revision artifacts for this camera-ready version.
> - We have carefully reviewed and revised our prior draft to be a clean manuscript without residual editing marks, duplicated wording, missing or double spaces, missing or double punctuation, etc.
> - We have converted the manuscript to follow the TMLR tex style of an accepted paper.
> - We have further revised the text and figure captions in the paper to consistently qualify the comparison between CLIQUE and SHAP, LIME, and ICI. The text is a bit more descriptive and a bit less comparative in assessing the information provided by these methods.
>
> Thank you again for choosing to accept our paper! If there are any further edits or changes needed prior to publishing, please let us know and we'll move quickly to accommodate.